# Energy and cost efficiency of Bitcoin mining endeavor

**Małgorzata Jabłczyńska[1]☯, Krzysztof Kosc[1]☯, Przemysław Ryś[1]☯, Paweł Sakowski📷[2]☯, Robert Ślepaczuk📷[2]☯*, Grzegorz Zakrzewski[2]☯**

**1** Faculty of Economic Sciences, Quantitative Finance Research Group, University of Warsaw, Warsaw, Poland, **2** Department of Quantitative Finance, Faculty of Economic Sciences, Quantitative Finance Research Group, University of Warsaw, Warsaw, Poland

☯ These authors contributed equally to this work.
* rslepaczuk@wne.uw.edu.pl

## Abstract

The main aim of the study is to analyze BTC mining's efficiency under current market conditions (December 2021), including soaring energy prices produced from many different sources in different geographical locations. After a thorough analysis of initial assumptions concerning the (1) price of mining machine with associated components and its effective amortization period, (2) difficulty and the hash rate of the BTC network, (3) BTC transaction fees, and (4) energy costs from various sources, we have found that currently, BTC mining is not profitable, except for some rare cases. The main reason for this phenomenon is the fast and unpredictable increase of difficulty of the BTC network over time which results in decreasing participation of already purchased mining machines in the BTC network hash rate. The research is augmented with a detailed sensitivity analysis of mining efficiency to initial parameters assumptions, which allows observing that the conditions for BTC mining to be efficient and profitable are very challenging.

## 1 Introduction

The main rationale for this paper is to address a collective misunderstanding in the matter of cryptocurrency mining efficiency which has become increasingly persistent in many sources like new mining initial coin offering (ICO) white papers, cryptocurrency news, blogs, online profitability calculators, etc. Our aim is to provide a meticulous analysis of the Bitcoin (BTC) mining efficiency when the rapid growth of mining difficulty and Bitcoin network hash rate are driving mining machines' participation rate (Mining machine participation rate is understood as the mining machine hash rate divided by the Bitcoin network total hash rate.) to erode in time quickly. We restrict our analysis of cryptocurrency mining efficiency solely to the Bitcoin network leaving all the other networks for further research. Nevertheless, our analysis and the methodology used can be generalized to other cryptocurrencies, which use proof-of-work and can be treated as a state-of-the-art approach to define when mining is efficiently taking into account all crucial characteristics of the given cryptocurrency.

**Data Availability Statement:** All relevant data are within the paper and its Supporting information files.

**Funding:** The author(s) received no specific funding for this work.

**Competing interests:** The authors have declared that no competing interests exist.

We refer to the issue of mining activity also from the macro-policy level addressing such important topics as the efficiency of the use of various energy sources and trying to answer the question of what could be the consequence of prolonging the unprofitability of mining activity for the whole ecosystem of cryptocurrencies. As far as we know, this is the first attempt to analyze the issue of BTC mining in such a thorough way. until now, the scientific literature does not provide us with reliable and complete models, with comprehensive "what-if" scenarios of mining activity. Additionally, we used the largest possible number of external variables which can affect the adequacy of our methodology and its potential for generalization for other cryptocurrencies.

In our work, we stated the main research hypothesis (RH1) as *Is BTC mining profitable under current mining conditions?* Additionally in order to refer to the details of mining efficiency we formulated the following research questions:

- RQ1: *What is the sensitivity of BTC mining profitability to the initial assumptions and future trajectories of main parameters?*

- RQ2: *Is there a strong positive relation between BTCUSD and the difficulty/hash rate of the BTC network?*

- RQ3: *What are the breakeven levels of the main BTC mining parameters in order to make this activity profitable?*

- RQ4: *What is the rational level of the amortization period for our mining machines?*

- RQ5: *What are the consequences of BTC mining efficiency for the future ability of blocks to be mined further?*

- RQ6: *Can we expect that blocks will not be confirmed if the efficiency of BTC mining is negative for an extended period of time?*

- RQ7: *Should we invest in BTC mining if we expect the BTC price to increase substantially?*

- RQ8: *Is it possible that technological advancements in the ASICs' performance could change the BTC mining landscape and make mining profitable?*

- RQ9: *What kind of new financial instruments may stimulate the BTC mining industry?*

We plan our paper as follows. In the second section, we present a literature review presenting several ICOs with the highest capitalization focused on cryptocurrency mining and publicly listed companies that announced that they undertook mining activity as well. Our model, its most important assumptions, and a detailed description of the data we used are presented in the third section. We provide a full description of our methodology and model in section number four. The fifth section states our main results. The one- and two-factor sensitivity analyses are presented in the sixth section. Section number seven describes the breakeven conditions for the BTC mining to be profitable, and the last section concludes.

## 2 Literature review and business mining activities review

### 2.1 Literature review

At the very beginning, we have to notice that papers describing the issue of the BTC (or any other cryptocurrency) mining efficiency in a detailed and rigorous scientific manner are very scarce. Thus, there is an urgent need to clarify this matter. Our paper tries to establish a groundwork for a wider discussion.

[1] presented a detailed analysis of bitcoin mining profitability. The calculations were based on mathematical formulae intended to objectively measure mining efficiency. This was an attempt to measure mining efficiency. However, there were no assumptions about the future evolution of the main drivers of their formulae.

[2] stated that one of Bitcoin's most significant problems is its seemingly insatiable use of electricity. In their research, along with providing the required power for Bitcoin (BTC) mining, a solid oxide fuel cell (SOFC) system fed either by natural gas or biogas as a renewable source of energy is used to supply the electricity demand. Thermodynamic modeling for the fuel cell system is applied to determine the required biogas or natural gas. For the proposed cases (grid-based, natural gas-fed SOFC, and biogas-fed SOFC), various scenarios depending on the Bitcoin price and mining difficulty were proposed. Also, the economic viability for each scenario in several countries was investigated and compared. Results indicate more profitability for grid-based mining in Bitcoin prices up to 20000 USD, but as the Bitcoin price increases SOFC based mining operations achieve reasonable profitability. It is shown that Iran, Russia, and China with cumulative cash flows of $87,300, $77,200, and $70,500 respectively, are the best countries to mine BTC using grid electricity while Iran, Canada, and Russia are the best countries using a natural gas-fed SOFC system.

[3] described the problem of energy consumption of cryptocurrency mining, however their study related digital coins other than Bitcoin. They ran experiments on the mining efficiency of nine kinds of cryptocurrencies and ten algorithms. Thereafter, the study provided an estimation of the global electricity consumption of the Monero mining activity and its carbon emission in China as a case study. They conclude, that although cryptocurrency mining and blockchain technology are promising, their influence on energy conversation and sustainable development should be further studied.

[4] estimated the environmental impact of Bitcoin mining and contributed to the discussion on the technology's supposedly large energy consumption and carbon footprint. This study applied the well-established Life Cycle Assessment methodology to an in-depth analysis of drivers of past and future environmental impacts of the Bitcoin mining network. It was found that, in 2018, the Bitcoin network consumed 31.29 TWh with a carbon footprint of 17.29 MtCO2-eq, an estimate that is at the lower end of the range of results from previous studies. The main drivers of such impact were found to be the geographical distribution of miners and the efficiency of the mining equipment. In contrast to the previous studies, it was found that the service life, production, and end-of-life of such equipment had only a minor contribution to the total impact and that while the overall hash rate is expected to increase, the energy consumption and environmental footprint per TH mined is expected to decrease.

[5] focused on Bitcoin price and its mining costs and showed that these two quantities are tightly interconnected and tend to a common long-term equilibrium. According to the Author, mining costs adjust to the cryptocurrency price with an adjustment time of several months up to a year. He wrote that recent developments suggest that we have arrived at a new era of Bitcoin mining where marginal (electricity) costs and mining efficiency play a prime role.

[6] focused on the NiceHash platform, which is the largest crypto-mining marketplace having a large number of active miners. In the NiceHash platform, sellers (also called, miners) mine coins to fulfill the selected buyers' orders. They proposed a system that can maximize the Bitcoin profit of miners by automatically selecting the most profitable mining algorithms and keeping track of the connection between miners and the mining pool.

[7] provided an updated estimation of the energy consumption of the Bitcoin network and a calculation of the evolution of the production cost of Bitcoin over time. They concluded that since June 2018 Bitcoin mining is no longer profitable for commodity miners without access

to electricity prices below 0.14 \$/kWh. This phenomenon explains why many Western miners have dropped out of the circuit, further increasing the centralization of mining activity in China. In addition, they estimated that the marginal cost of the production of Bitcoin is around 1952 USD. Below this price, the cost of mining would not be profitable, even with the most efficient equipment and the lowest possible price for the energy required. According to the Authors, this could lead to a massive exit of the biggest mining players, with unpredictable consequences for the future of this cryptocurrency.

In order to create incentives for early investment, [8] proposed that wind farm investors could hedge electricity price risk by simultaneously investing in a cryptocurrency mining facility that uses electricity as input to produce newly minted Bitcoins. As electricity and Bitcoin prices are uncorrelated, the ability to switch between these outputs allows the wind farm to maximize revenues and minimize losses. They develop a numerical application under the real options approach to determine the financial impact of the investment in a Bitcoin facility for the wind energy producer that would allow it to optimally switch outputs depending on the relative future prices of electricity and Bitcoins. The short-term energy price and Bitcoin price/mining-difficulty ratio were modeled as distinct stochastic diffusion processes. The results indicated that the option to switch outputs significantly increases the generator's revenue while simultaneously decreasing the risk of anticipating the construction.

[9] focused on the bitcoin miners as they play an important role in the proof-of-work consensus mechanism of bitcoin to create trust in the currency. Miners offer their services against a reward while recurring expenses. Their results show that bitcoin mining has become less profitable over time to the extent that profits seem to converge to zero. They analyzed the actors involved in the bitcoin system as well as the value flows between these actors using the e3value methodology. The value flows were quantified using publicly available data about the bitcoin network. Using estimates of hardware investments and expenses for electrical power, they can calculate the expenses the miner should have. At the end of the analysis period, the marginal profit of mining a bitcoin becomes negative, i.e., to a loss for the miners. This loss is caused by the consensus mechanism of the bitcoin protocol, which requires a substantial investment in hardware and significant recurring daily expenses for energy. Therefore, a sustainable cryptocurrency needs higher payments for miners or more energy-efficient algorithms to achieve consensus in a network about the truth of the distributed ledger.

[10] focused on computational power demand during the proof-of-work process rather than estimating the whole energy intensity of Bitcoin mining. They made use of Bitcoin blockchain data to estimate the energy consumption and power demand of Bitcoin mining. They considered the performance of 269 different hardware models (CPU, GPU, FPGA, and ASIC). For estimations, they defined two metrics, namely; minimum consumption and maximum consumption. The targeted time span for the analysis was from 3 January 2009 to 5 June 2018. They showed that during this period the historical peak of power consumption of Bitcoin mining took place during the bi-weekly period commencing on 18 December 2017 with a demand of between 1.3 and 14.8 GW. This maximum demand figure was between the installed capacities of Finland (16 GW) and Denmark (14 GW). They also showed that, during June 2018, the energy consumption of Bitcoin mining from difficulty recalculation was between 15.47 and 50.24 TWh per year.

One of the most recent research about Bitcoin mining efficiency was presented by [11]. The authors came to the conclusion that mining activity is profitable (as of the publishing date) but only under certain assumptions. The authors also concluded that the Bitcoin mining network is mainly powered by renewable energy, dominated by hydro sources.

[12], proposed a modification of a transaction fee scheme and recommended that components of a fee should be associated with resources utilized: network, computation, or storage.

This is an interesting direction of consideration for blockchain developers, especially now, when the moment of block reward reduction is imminent and the importance of the level of future BTC transaction fees will naturally grow.

A recent research by [13] presented the convergence effect of BTC prices and its marginal cost of production. The conclusion has been confirmed using simple OLS regression as well as the VAR model. It should be underlined that no hardware costs (i.e. capital expenditure (CAPEX)) were taken into account in the analysis. Therefore, one can not properly conclude on actual profitability of mining based on the final results of this paper.

[14] presented the thesis explaining BTC price dynamics in late 2017 and early 2018. The increase and following sharp decrease in prices was related to the speculative dynamics that were able to materialize after the introduction of futures on the CME and CBOE.

[15] described the evolution of mining hardware over time, which is one of the most important drivers of mining profitability. He calculated daily Bitcoin revenue in dollars, per gigahash per second (GH/s) of mining performance, over time.

A more realistic approach, in our opinion, was taken by [16]. The variation of mining difficulty over time was explicitly assumed in a form of linear regression of log-transformed raw variable. Thus, it presents the importance of the dynamics of this part of the model which is omitted by naive profitability calculators available on many websites, as Deutsch underlines. Such flaws in models might lead to substantially biased conclusions and eventually to severe losses for individuals who relied on one of the faulty mining calculators. Deutsch presented sensitivity analyses of mining profitability versus a synthetic measure alpha. In our research, an attempt is made to analyze sensitivity versus the most basic components of profitability. It is also worth mentioning that Deutsch extended his work with the block-reward halving-days effect.

There were also some other papers, indirectly referring to mining activity through analysis of energy shocks and their co-movements with carbon market dynamics ([17]) or by referring to the causality between CPU and traditional energy and green markets ([18]).

The presented literature review shows that there were some attempts to the issue of mining activity and its profitability but they only partially referred to the main subject or were published a quite long time ago. Moreover, none of these works covered the subject of BTC mining efficiency thoroughly enough in order to refer to all our hypotheses and research questions and this was the reason why we decided to undertake this topic. The issues which require further investigation are as follows: detailed sensitivity analysis of the complex profitability of BTC mining, linking the characteristics describing the BTC blockchain with mining activity, and the use of various energy sources in this process.

## 2.2 Review of mining activities financed by ICO

Direct information about the profitability of real mining enterprises is very limited. Companies related to the mining industry refrain from publishing any detailed information on efficiency or any other aspect of their businesses. It is virtually impossible to obtain any precise financial statements of such companies and scraps of information present on their websites are very far from being a reliable source of scientific data. In this part, we analyze the business profitability and potential prospects of the ICO-financed mining companies. It was assumed that the ICO exchange market is characterized by some level of informational efficiency. Even though the Efficient Market Hypothesis ([19]) could be difficult to defend for the ICO market, we assume that general conclusions drawn from ICO prices evolution are justifiable.

i. Envion, 2018

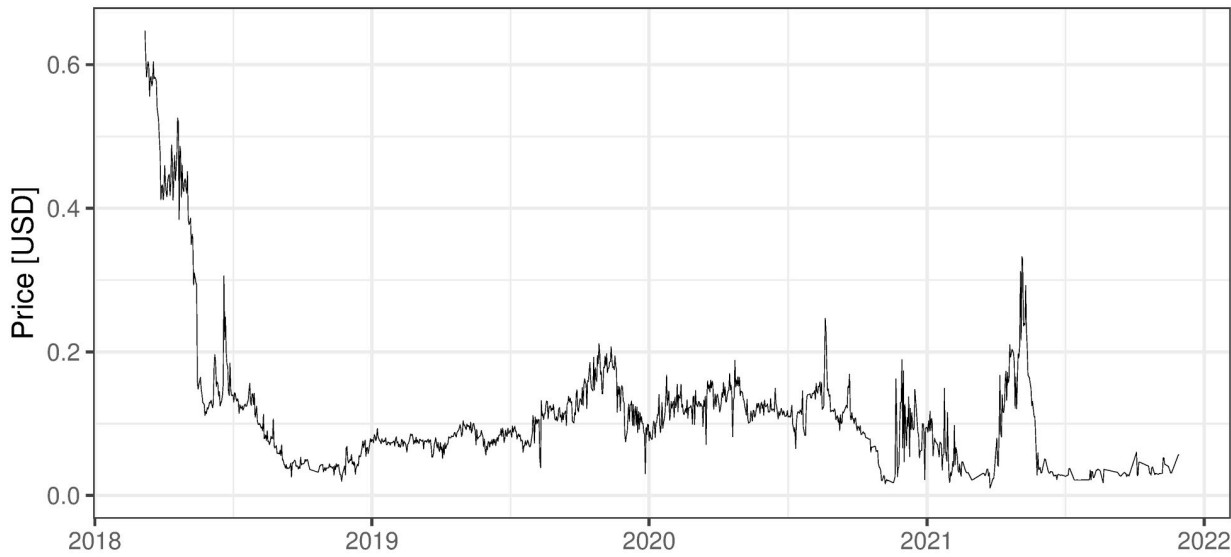

**Fig 1. Envion token price in USD.** Source: https://coinpaprika.com/coin/evn-envion/ (downloaded at 2021/12/03).

One of the biggest attempts to create a cryptocurrency mining enterprise is Swiss Envion. Their main advantages presented on the website (https://www.envion.org/en/) are: the mobility of the infrastructure, the cheap energy sources (mainly from solar sources and wind farms), remote management, and patented cooling technology. The company was able to attract substantial investment capital. The ICO finished with 100m USD raised within a short period of time i.e. from 2017/12/15 until 2018/01/15. Based on the White Paper one might conclude that the primary activity was Ethereum mining. The White Paper described it as a so-called smart mining operations approach. This means that the company was prepared to flexibly switch from mining one cryptocoin to another. The operation has been performing only for a few months up to date, nevertheless, the most objective conclusions about its profitability can be drawn from Envion ICO quotes. Fig 1 presents a quotation from the HitBTC exchange.

Based on the real valuation of the ICO one may conclude that the prospects of this mining activity may not match the initial ICO valuation. Since the first day of trading, Envion has lost about 86% of the ICO nominal price of the token (as of 2021/12/03).

ii. Ambit, 2018

The other mining company, established in 2018, and financed via ICO in June 2018, was Ambit with headquarters in Georgia. No detailed public data was available for this company except for the fact that fundraising was closed. The price of the token was defined at 0.5 USD and 88m tokens were planned to be sold. Final fundraising has been closed with 4m USD (https://icobench.com/ico/ambit as of 2018/07/25). The company promised very high profits, exceeding 86% annually (https://ambitmining.io/ as of 2018/07/23) but unfortunately, we did not have any detailed information about current ICO quotes nor revenues of the company from mining.

iii. Veritas and Xenium, 2018

The other ICO-funded mining companies were Veritas Mining and Xenium, both related to each other, that issued respectively VRTM and XENM tokens. We are not able to evaluate

the evolution of token prices as one of them is not listed as of the research date and the quotations of the second one have hardly any liquidity.

iv. Other ICO

There were many other cases of mining ICO gathering funds mainly in 2017 and 2018 but many of them did not develop after their initial fundraising stage and were not able to gather the required funds to start their operations or went bankrupt shortly after successful or unsuccessful ICO.

## 2.3 Other mining activities, non ICO-financed

There are also several other companies, non-ICO-financed, that are oriented on mining activities. They are financed in the more traditional way and usually traded on stock exchanges, which for sure increases their credibility and maybe affects the standard of their management. Five examples of such companies are presented below with the evolution of their shares' prices (Figs 2 and 6). First two companies: 360 Blockchain and HashChain Technology (Figs 2–3) were able to bring their share prices down to around zero. The other three examples are DMG, HIVE Blockchain, and Neptune Dash (Figs 4–6), all quoted on TSE, behaved a little bit better, but analyzing their prices we can see that they were almost bankrupt at the end of 2020 and only sharp move-up in Bitcoin prices rescued them before this event. Taking the above into account and partly referring to RQ7 we can deliberate if the investment in mining companies is not only a pure play on BTC prices and if it is like this whether should place such a bet through some Bitcoin mining enterprises instead of just pure investment in financial instruments tracking Bitcoin prices (ETFs, futures, options, etc.).

i. 360 Blockchain (CSE:CODE). The company is traded on Canadian Securities Exchange. The current share price is at the level of 0.125 CAD (data as of 2021/12/03).

ii. HashChain Technology (TSXV:KASH). The company is traded on Toronto Stock Exchange. The current share price is at the level of 0.015 USD (data as of 2021/12/03).

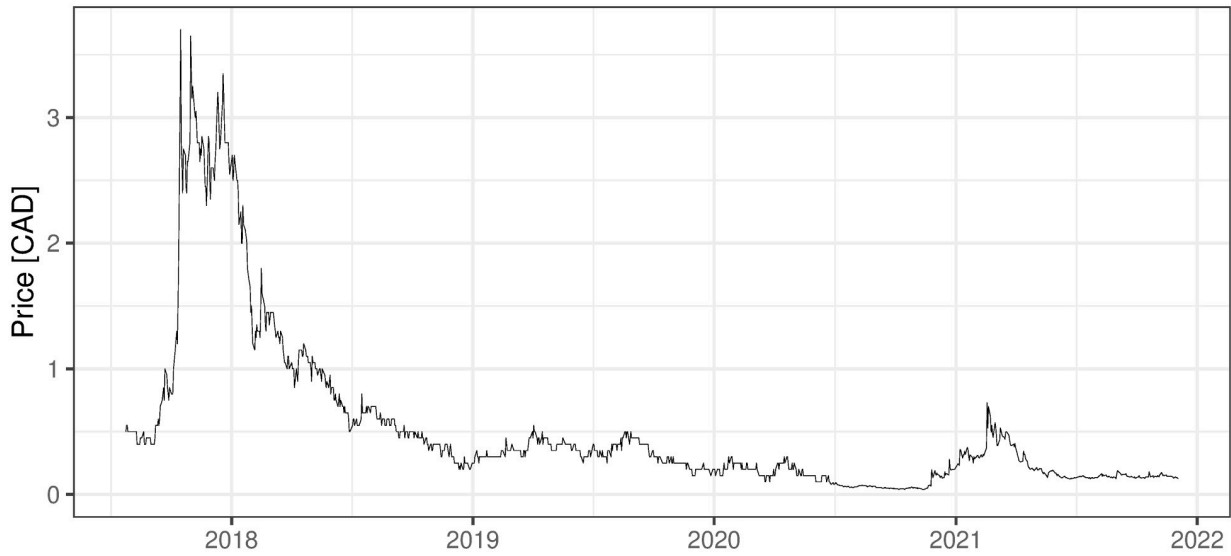

**Fig 2. The daily fluctuations of 360 Blockchain prices on CSE.** Source: https://finance.yahoo.com/quote/CODE.CN/history?period1= 1469836800%20period2=1638489600%20interval=1d%20filter=history%20frequency=1d%20includeAdjustedClose=true (downloaded at 2021/ 12/03).

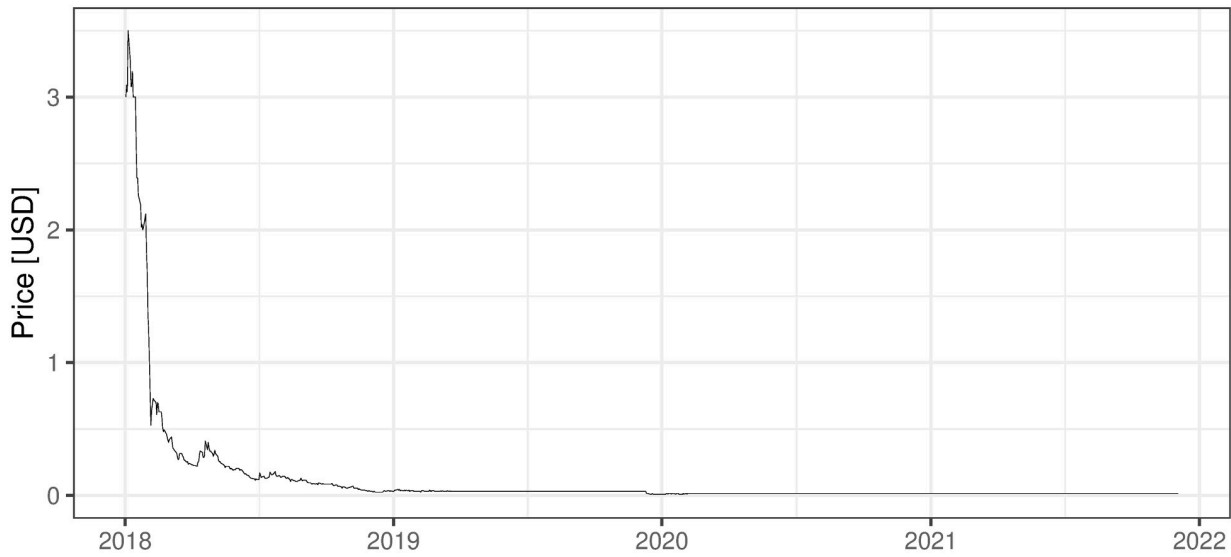

**Fig 3. The daily fluctuations of HashChain Technology prices on TSE.** Source: https://ih.advfn.com/stock-market/TSXV/KASH/historical/more-historical-data (downloaded on 2021/13/03).

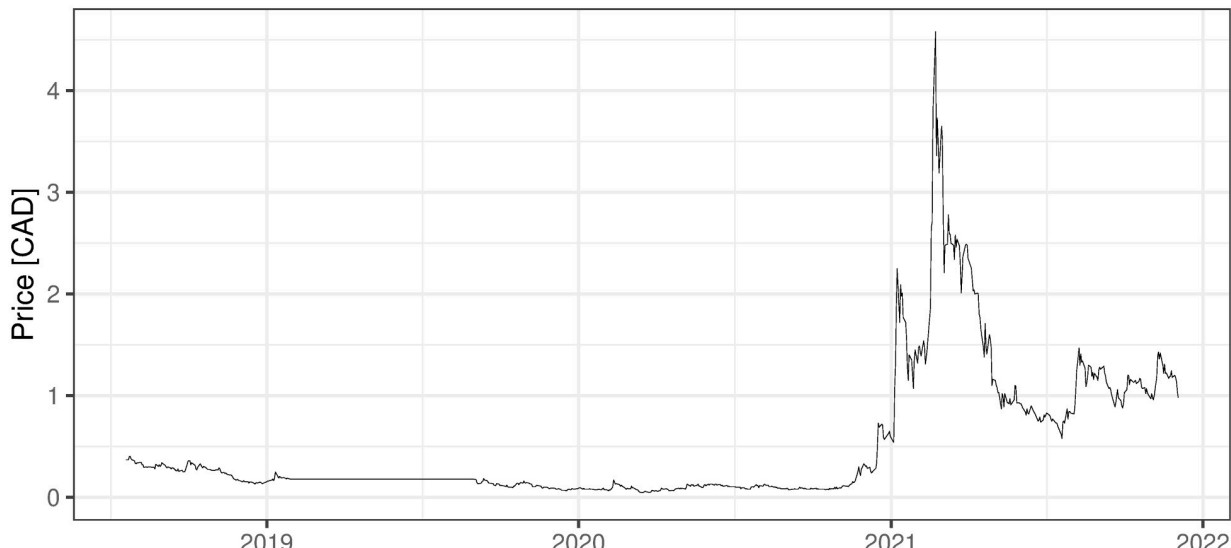

**Fig 4. The daily fluctuations of DMG Blockchain prices on TSE.** Source: https://finance.yahoo.com/quote/DMGI.V/history?p=DMGI.V (downloaded on 2021/12/03).

iii. DMG Blockchain (TSXV:DMGI). The company is traded on Toronto Stock Exchange. The current share price is at the level of 0.98 CAD (data as of 2021/12/03).

iv. HIVE Blockchain (TSXV:HIVE). The company is traded on Toronto Stock Exchange. The current share price is at the level of 4 USD (data as of 2021/12/03).

v. Neptune Dash (TSXV: DASH). The company is traded on Toronto Stock Exchange. The current share price is at the level of 0.59 CAD (data as of 2021/12/03).

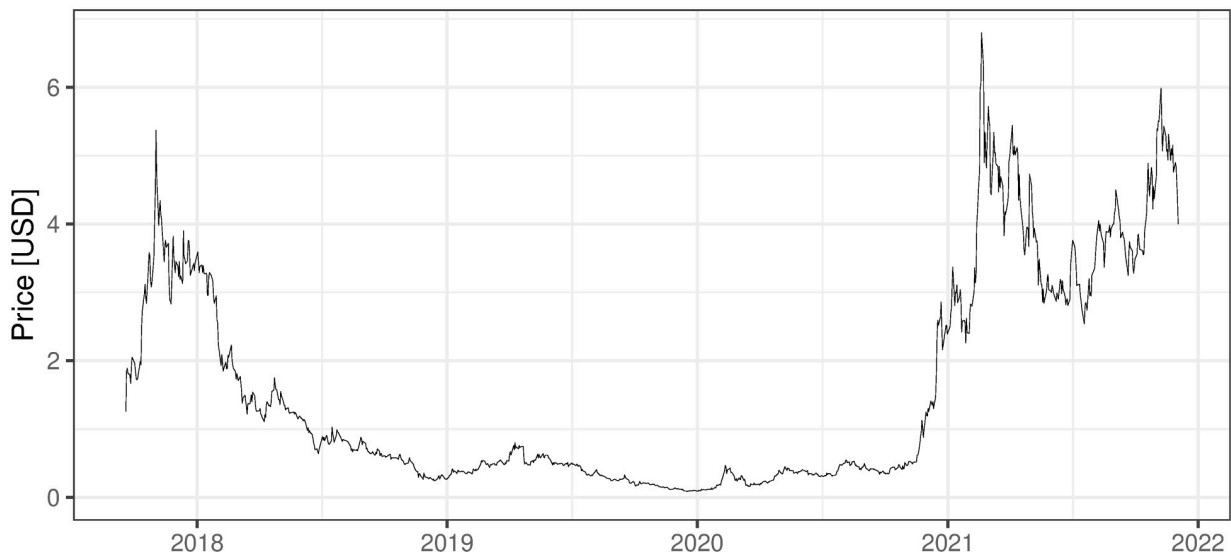

**Fig 5. The daily fluctuations of HIVE Blockchain prices on TSE.** Source: https://finance.yahoo.com/quote/HIVE.V/chart?p=HIVE.V (downloaded on 2021/12/03).

vi. Argo Blockchain. An interesting initiative in the industry is created by the company Argo Blockchain (https://www.argomining.co) (Fig 7). Their aim is to create an opportunity to invest in cryptocurrency mining via the so-called "mining-as-a-service" (MaaS). Argo Blockchain organized the IPO on London Stock Exchange and is a purely crypto-mining company listed on LSE.

vii. Apart from the ICO-founded mining companies and stock exchange trading ones, there are several other forms of attracting capital to the mining industry, including a particularly

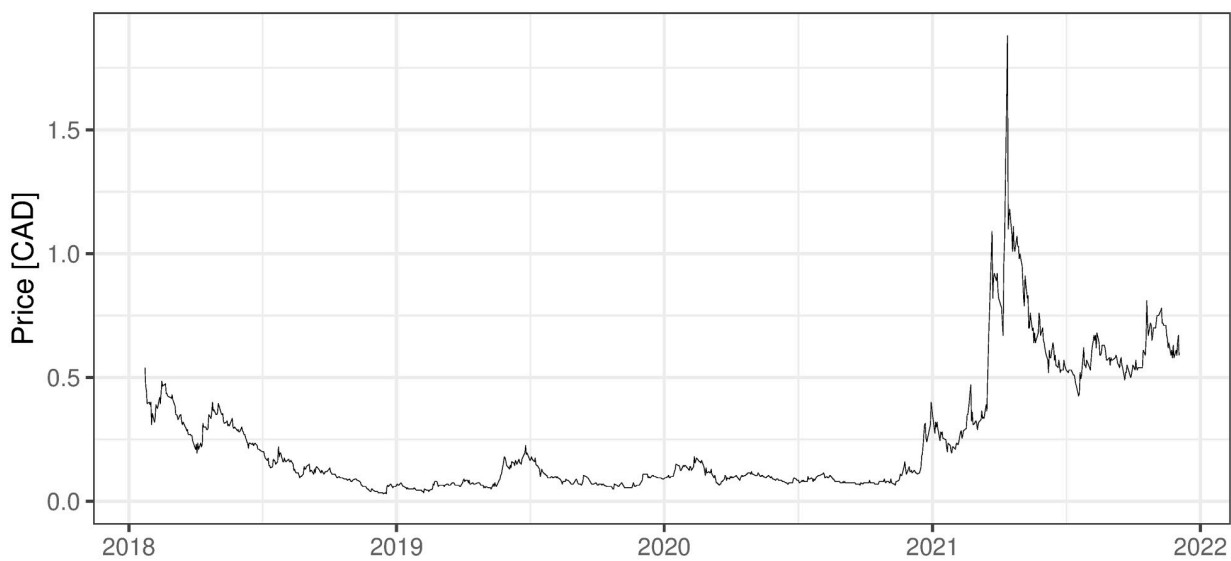

**Fig 6. The daily fluctuations of Neptune Dash prices on TSE.** Source: https://finance.yahoo.com/chart/NDA.V (downloaded on 2021/12/03).

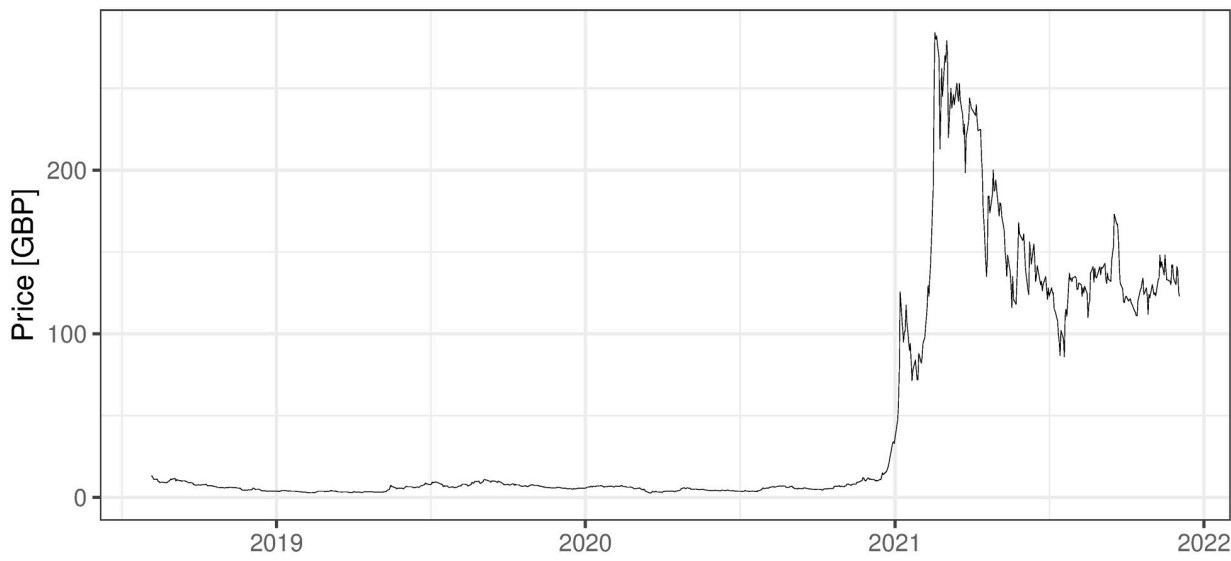

**Fig 7. The daily fluctuations of Argo Blockchain prices on LSE.** Source: https://query1.finance.yahoo.com/v7/finance/download/ARB.L?period1=1533340800&period2=1639180800&interval=1d&events=history&includeAdjustedClose=true (downloaded on 2021/12/03).

interesting one heading from Poland. The 'Pracownia Finansowa sp. z o.o.' via one of its daughter companies 'Pracownia Nowych Technologii sp. z o.o.' collected capital via limited partnership company structure where investors play a limited partner role. The company stated in 2017 and 2018 that cryptocurrencies mining is one of the was profitable and fastest growing businesses (https://www.pnt24.pl/oferta/inwestycja-w-kopalnie-btceth as of 2018–07-21). It also presented that the mining profitability ranges from 50% to 150% annually (https://www.pnt24.pl/oferta/inwestycja-w-kopalnie-btceth as of 2018–07-21) what in the case of BTC mining was rather in strict contradiction with our main results showing that in 2018 ([20]) and currently, mining was not profitable. The certain confirmation of our results may be the fact that PNT sp. z o.o. does not exist anymore and the whole project ended in bankruptcy.

## 3 Crucial assumptions and data

Our methodology is based on thorough efficiency analysis and takes into account current conditions of BTC mining (as of 2021/12/03) in the main worldwide jurisdiction. In order to perform such an analysis we have to make the following assumptions about our crucial parameters and clearly state our sources. The following analysis covers all this information. The metrics introduced in this section (section 3) describe the state of the BTC blockchain affecting the profitability of mining activity and then they are used in the Methodology section (section 4) presenting the model evaluating the profitability of BTC mining. We visualize these metrics in this section (section 3) in order to give the reader the intuition about their past fluctuations and make it possible to understand the dynamics of the BTC network. Thus, this section (section 3) constitutes the introduction to the methodological part presented in the Methodology section (section 4). The division of methodology in these two parts is crucial from the point of view of potential understanding of the whole dynamics of the variables' relations described by the equations from (6) to (18).

## 3.1 BTC network hash rate

The monthly change of the BTC network hash rate in the future will be on the level of its historical trajectory. In practice we assume one year average of monthly BTC hash rate change (1Y avg of monthly BTC hash rate change%) (Data source: https://api.blockchain.info/charts/hash-rate (downloaded at 2021/12/03)).

## 3.2 BTC network difficulty

BTC network difficulty is strictly connected with BTC network hash rate according to the following formula:

$$D = \frac{T \cdot HR}{2^{32}} \tag{1}$$

where: $D$—stands for BTC network difficulty, $T$—stands for the time of block mining, which is set to be about 10 minutes according to BTC white paper (https://bitcoin.org/bitcoin.pdf as of date 2021/12/03), $HR$—means BTC network hash rate.

Short-term departures from this equilibrium cause the difficulty of the BTC network to be adjusted in such a way that the required time of mining a single block will last the mandatory 10 minutes once again. The adjustment is made every 2016 blocks (https://en.bitcoin.it/wiki/Difficulty as of date 2021/12/03), i.e. approximately every two weeks (Data source: https://charts.coinmetrics.io/network-data/ (downloaded at 2021/12/03)).

## 3.3 The mining machine—Antminer S19J Pro

We assumed that Antminer S19J Pro produced by Bitmain (https://www.bitmain.com/) is currently the most efficient and widely used BTC mining machine, and for sure much better than its predecessor Antminer S19 used widely in 2017, 2018, and 2019. We take into account the combination of its price, amortization period, hash rate, and energy consumption during the mining process. We would like to stress that it does not mean that there are no other mining machines (http://bit.ly/41ju4e0 as of date 2021/12/03) that are more efficient with regard to single characteristics describing the mining efficiency (hash rate, energy consumption or price), but we have just selected the most popular and efficient taking all these parameters simultaneously into account. It additionally means that if we had performed an additional level of sensitivity analysis only with regard to various types of BTC mining machines then the profitability of BTV mining would be at best lower than that reported in this study.

i. Price + Taxes + Shipment costs

- The price of Antminer S19J Pro is assumed on the level of $1.2064 \times 10^4$ USD (https://bitmain.com/ as of 2021/12/03).

- PSU (power supply unit) is assumed on the level of 160 USD (https://bitmain.com/ as of 2021/12/03).

- Necessary taxes or shipment costs increase the price of single mining machine (e.g. VAT, etc.) are assumed on the level of: 23%.

ii. Amortization

The amortization period was assumed to be at the level of 12 months. We are aware of the fact that in some cases when technological advancements occur faster, the amortization period could be shorter. Taking into account that during the last year, there was no revolutionary change within the spectrum of the most efficient mining machines, we decided to keep 12

months to fix the attention as it is the sufficiently precise assumption for the purpose of this research.

The other approach to the amortization period is to make this period conditional on the time which passes until the moment our mining machine will not be able to cover the sum of its monthly energy expenses and maintenance costs after the cost of buying this hardware have already been incurred. A detailed analysis of this feature will be presented in the Results section (Section 5).

iii. Hash rate

The hash rate of Antminer S19J Pro, according to its producer, is 104 TH/s (http://bit.ly/3SAYeWd as of 2021/12/03).

iv. Energy consumption

The power consumption of a single mining machine is 3068 W per hour (http://bit.ly/3KtHhe0 as of 2021/12/03).

### 3.4 Unit energy price

The average price of energy per kWh for the institutional investor in the most common jurisdiction, i.e. China, is 0.09 USD. Additionally, we take into account three other levels of unit energy costs: US (0.15), Poland (0.19), and minimum available (0.01), which currently can be obtained in Iran (Data source: http://bit.ly/41gSGE5 (downloaded at 2021/12/03)).

### 3.5 Maintenance cost

We assumed that maintenance should cover all costs connected with operational and logistic issues, e.g. paying the rent, setting up the machine, its everyday maintenance, and especially the cost of all periods when it is stopped due to hardware failures, overheating, etc. The key components affecting this cost category are periods of downtime of our mining machine, therefore this cost is expressed as a percentage of the value of BTC mined per year. This cost was assumed at the level of 25% of the value of BTC mined per year.

### 3.6 BTCUSD price

In the base case scenario we assumed the price of $5.3671616 \times 10^4$ USD for 1 BTC (Data source: https://charts.coinmetrics.io/network-data/ (downloaded at 2021/12/03)).

### 3.7 BTC transactions per day

This variable refers to the number of all BTC transactions which occurred in all blocks during that day (Data source: https://charts.coinmetrics.io/network-data/ (downloaded at 2021/12/03)).

### 3.8 BTC mined per day

This variable refers to the number of new coins that have been brought into existence on that day (Data source: https://charts.coinmetrics.io/network-data/ (downloaded at 2021/12/03)).

### 3.9 BTC transaction fees per day

This variable refers to the value of all BTC fees paid in the network on the given day (Data source: https://charts.coinmetrics.io/network-data/ (downloaded at 2021/12/03)). BTC fee should be understood as the total value of BTC paid by the participant using the BTC network

in order to perform various activities connected mostly with BTC transfers and confirmations. These fees are earned by the "miners" and they increase the profitability of the mining process. Most often we use this variable in percentage terms, dividing "BTC transaction fees per day" by "BTC mined per day".

### 3.10 BTC blocks per day

This variable refers to the number of blocks which were mined per day (Data source: https://charts.coinmetrics.io/network-data/ (downloaded on 2021/12/03)).

### 3.11 Theoretical number of BTC blocks mined per hour

Based on the seminal paper of [21] we assume that blocks are mined every 10 minutes what means that there are 6 blocks per hour.

### 3.12 Sensitivity analysis

The evolution of mining profitability with regard to a given factor or factors is presented and analyzed ceteris paribus all other factors.

### 3.13 Breakeven analysis

We made the same assumption as in the sensitivity analysis section.

### 3.14 Additional variables

Based on the raw data described above we created the following new variables:

i. Monthly BTC hash rate change%

$$\text{Monthly BTC hash rate change}\% = \frac{\text{BTC network hash rate on the day } t}{\text{BTC network hash rate on the day } t-31} - 1 \ (2)$$

ii. Monthly BTC difficulty change%

$$\text{Monthly BTC difficulty change}\% = \frac{\text{BTC network difficulty on the day } t}{\text{BTC network difficulty on the day } t-31} - 1 \ (3)$$

iii. BTC transactions per block

$$\text{BTC transactions per block} = \frac{\text{BTC transactions per day}}{\text{BTC blocks per day}} \qquad (4)$$

iv. Fees%—daily fees as% of daily block rewards.

$$\text{Fees}\% = \frac{\text{BTC transaction fees per day}}{\text{BTC mined per day}} \qquad (5)$$

**Table 1. Descriptive statistics of the main important data.**

| | BTCUSD | BTC network hash rate [TH/s] | Monthly BTC hash rate change% | BTC BTC network difficulty | BTC mined per day [BTC] | BTC blocks per day | BTC trans. fees per day [BTC] | BTC trans. per block | Fees% |
|---|---|---|---|---|---|---|---|---|---|
| min | 66.34 | 67 | -60.55 | 1.01e+07 | 362.50 | 58.00 | 7.37 | 165.28 | 0.18 |
| Q1 | 442.66 | 367769 | 0.49 | 4.94e+10 | 1675.00 | 141.00 | 17.86 | 779.89 | 0.73 |
| avg | 8752.52 | 42194334 | 19.08 | 5.84e+12 | 2481.99 | 152.44 | 75.79 | 1450.84 | 4.61 |
| Q2 | 3392.56 | 6578538 | 12.25 | 8.88e+11 | 1975.00 | 151.00 | 34.70 | 1580.06 | 1.47 |
| Q3 | 9244.23 | 87249038 | 28.51 | 1.28e+13 | 3725.00 | 163.00 | 81.78 | 2055.60 | 5.76 |
| max | 67541.76 | 198624496 | 240.04 | 2.50e+13 | 6500.00 | 260.00 | 1495.75 | 2767.47 | 77.20 |
| std dev | 14486.74 | 53987199 | 31.65 | 7.49e+12 | 1258.96 | 19.19 | 112.63 | 702.80 | 6.96 |
| last year avg | 45416.03 | 142352848 | 3.64 | 2.00e+13 | 901.23 | 144.20 | 65.07 | 1899.94 | 7.47 |
| last qtr avg | 55090.73 | 150319651 | 10.93 | 2.04e+13 | 931.66 | 149.07 | 14.21 | 1810.32 | 1.54 |

Note: The time period for this data is between 2013/05/01 and 2021/12/03.

Table 1 presents the main descriptive statistics of our data downloaded in the form of time series. All our crucial assumptions set in order to present the final calculations can be found in Table 2 in Methodology section (Section 4).

As presented in Table 1, the maximum BTCUSD level was 67541.76. Average monthly BTC hash rate change% is at the level of 3.641% and 10.93% respectively for the last year and the last quarter. The shorter average (last quarter) shows higher values, however, one should remember that the shorter the average period the more volatile the average is. Nevertheless, this higher value of BTC monthly hash rate change% may be regarded as a sign of the fact that mining is once again more attractive after a substantial increase in BTCUSD price. The average number of BTC mined per day reached 901.233 and 931.662 for the last year and the last quarter respectively. Theoretically, there should be 900 BTC generated as a block reward. The gap between theoretical and realized values is produced by the fact that hash rate increase is close to a continuous process while difficulty adjustment is a strictly discrete process. Therefore, there is a lag between the real-time period between consecutive block confirmation and difficulty target adjustment.

Fig 8 shows correlations between analyzed variables. Overall, we can distinguish groups of variables with very strong positive, and negative correlations and with no correlations. We notice strong negative correlations between such variables like *generatedCoins, BTCBlockReward, BTCMined12M*, and *transactionsPerBlock12M*, *transactionsPerBlock*. Positive correlations can be noticed between such pairs as *BTCUSD* and *hashrate12M* or *averageDifficulty* and *hashrate12M*. At the same time, we can see that *BTCUSDChange1W* and *dailyReturnsBTC* is not correlated with other variables used in this study.

In the remaining part of this section, graphical fluctuations of our main time series are presented in order to show the main trends in our data and their dynamics.

As can be seen in Fig 9 BTCUSD price remained below 1000 USD until the end of 2016 with the very volatile period in late 2013 and early 2014. The cryptocurrency gained popularity in late 2016 and a strong increasing trend started in 2017. Late 2017 was characterized by enormous hype on BTC which lifted its price to the level of 20000 USD. Since the beginning of 2018, there was a significant correction of the upward trend but BTC did not go below 10000 USD until March 2018. The 2017/2018 peak is by some researchers attributed to the introduction of futures on BTCUSD on CBOE and CME (Hale et al. 2018) and the entering of the

**Table 2. Summary of all our crucial assumptions, types of data, and data sources.**

| Single assumption | unit | Final value | Type of data | Data source |
|---|---|---|---|---|
| Mining machine net price (Antminer S9) | USD | 12064 | static | https://shop.bitmain.com/ |
| PSU net price | USD | 160 | static | https://shop.bitmain.com/ |
| Taxes&Shipment% | % | 23 | static | Internal assumption |
| Amortization period | months | 12 | static | Internal assumption |
| Hardware hash rate (Antminer S9) | TH/s | 104 | static | https://shop.bitmain.com/ and https://99bitcoins.com/antminer-s9-review/ |
| Hardware power consumption (Antminer S9) | W | 3068 | static | https://shop.bitmain.com/ and https://99bitcoins.com/antminer-s9-review/ |
| Minimal unit energy price | USD/kWh | 0.01 | static | other sources[b] |
| China unit energy price | USD/kWh | 0.09[a] | static | http://bit.ly/41gSGE5 |
| Poland unit energy price | USD/kWh | 0.19 | static | http://bit.ly/41gSGE5 |
| US unit energy price | USD/kWh | 0.15 | static | http://bit.ly/41gSGE5 |
| Maintenance cost | % of total value of BTC mined per year | 25 | static | Internal assumption |
| BTCUSD | USD | 53671.6 | Time series | https://charts.coinmetrics.io/network-data/ |
| BTC network hash rate | TH/s | 1.87e+08 | Time series | https://api.blockchain.info/charts/hash-rate |
| 1Y avg of monthly BTC hash rate change% | % | 3.64 | Time series | Own calculations |
| BTC network difficulty | TH/s | 2.23e+13 | Time series | https://charts.coinmetrics.io/network-data/ |
| BTC transactions per day | number | 289452 | Time series | https://charts.coinmetrics.io/network-data/ |
| BTC mined per day | BTC | 1050 | Time series | https://charts.coinmetrics.io/network-data/ |
| BTC transaction fees per day | BTC | 13.39 | Time series | https://charts.coinmetrics.io/network-data/ |
| BTC blocks per day | number | 168 | Time series | https://charts.coinmetrics.io/network-data/ |
| Theoretical BTC blocks mined per hour | number | 6 | static | https://bitcoin.org/bitcoin.pdf |
| BTC transactions per block | number | 1722.9 | Time series | Own calculations |
| Fees% | % | 1.3 | Time series | Own calculations |
| 1Y avg of Fees% | % | 7.47 | Time series | Own calculations |
| BTC block reward | BTC | 6.25 | static | https://www.blockchain.com |
| Theoretical BTC mined per year | BTC | 328500 | static | Own calculations |

[a] This is our base case scenario.

[b] We have numerous sources on unit energy prices: https://www.forbes.com/sites/christopherhelman/2018/01/16/bitcoin-mining-uses-as-much-power-as-ireland-and-why-thats-not-a-problem/ https://cryptocurrencynews.com/daily-news/mining/cheapest-places-mining-bitcoin/, https://www.howtogeek.com/349033/why-it%E2%80%99s-nearly-impossible-to-make-money-mining-bitcoin/, https://ca.reuters.com/article/businessNews/idCAKBN1F10BU-OCABS/

market by at least some institutional investors. However, in our opinion, the main and quite obvious reason for the market correction observed in the first half of 2018 was a normal adjustment of prices after an abnormal departure from the long-term trend, characterized by very volatile turmoils. After some normalization of prices in 2018–2020, the prices started to sharply move up and reached the maximum in the first quarter of 2021 at the level above 65000 USD for 1 BTC. Then BTCUSD price went down more than 50%, to 30000 USD per 1 BTC, but finally rebounded more than 100% reaching a new maximum price at the level of 67500 USD per 1 BTC in late 2021.

Fig 10 shows that the BTC network hash rate has been in an increasing trend almost all the time besides mid-2021 when rumors concerning the ban of crypto mining in China became a fact. In mid-2021 one can see an exceptional drop in the BTC network hash rate reaching

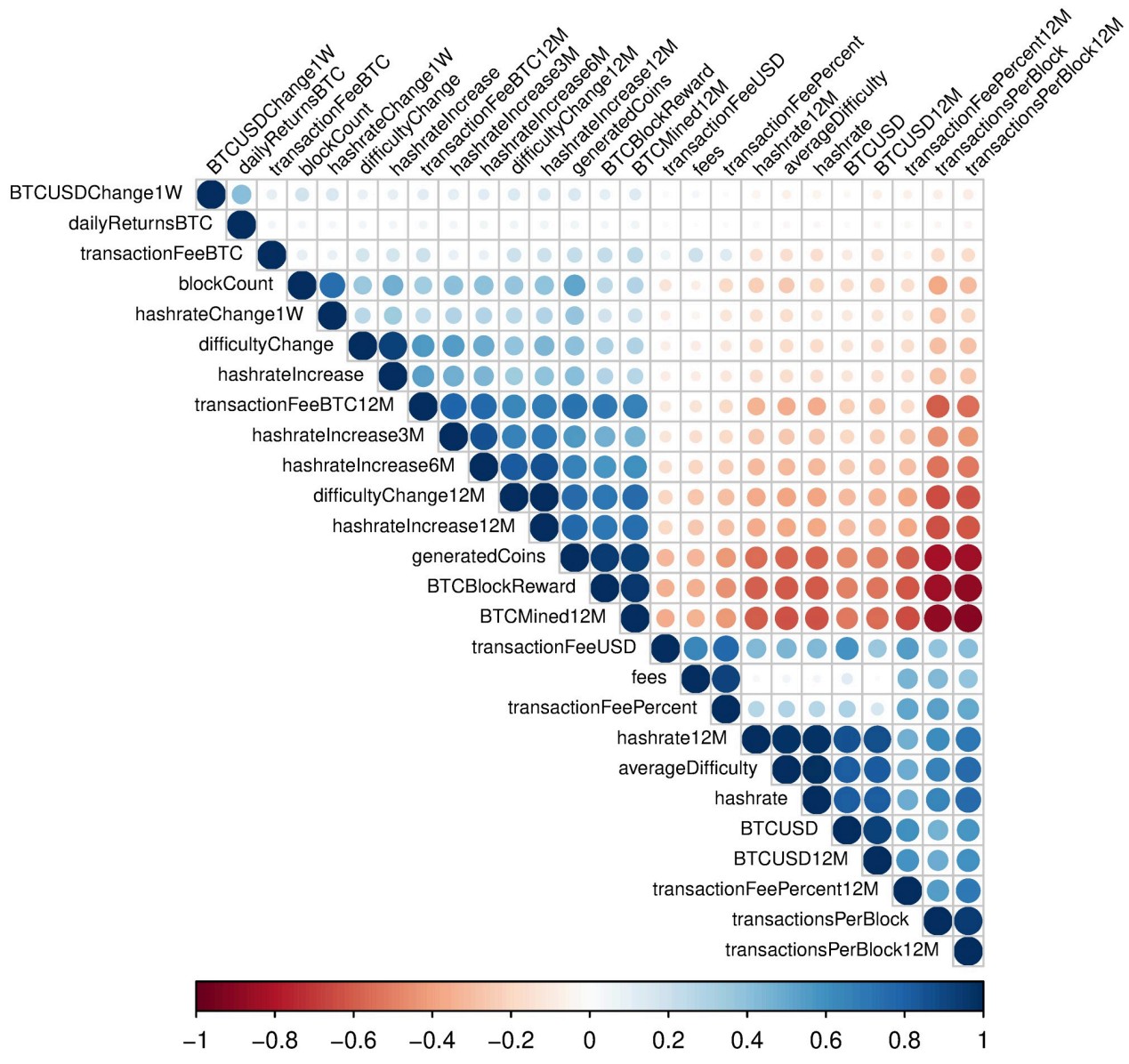

**Fig 8. Correlations of the analyzed variables.** The plot presents Pearson correlation coefficients among analyzed variables. Blue and red colors denote positive and negative correlations, respectively. The size of the circles is proportional to the absolute value of correlations. The time period for this data is between 2013/05/01 and 2021/12/03.

almost 75% and currently after several months, the BTC network hash rate is approaching the previous maximum at the level of 2.0e+08. Other exceptional periods with decreasing hash rates occurred historically when the BTC market was less mature e.g. August 2011 to October 2012. Nevertheless, in the most recent history of BTC, i.e. since 2014, the network hash rate developed close to exponentially. One can observe this phenomenon when plotting the hash rate with a logarithmic scale.

The long-term increase in the BTC network hash rate is confirmed by the analysis of monthly BTC hash rate change (calculated according to formula 2) which oscillates around 19% on average for the whole period and 3.6% for the last year. In order to analyze long-term

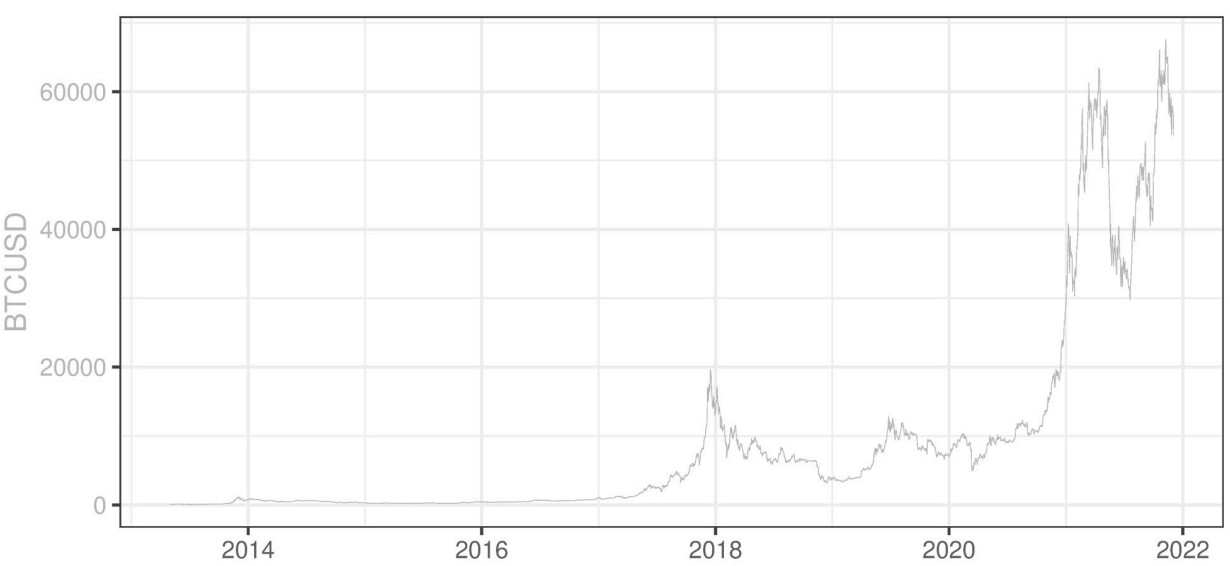

**Fig 9. BTCUSD historical prices.** Note: Data covers the period between 2013/05/01 to 2021/12/03. Source: https://charts.coinmetrics.io/network-data/.

trends the 1-year average of monthly BTC hash rate change% (later on referred to as 1Y avg of monthly BTC hash rate change%) was plotted alongside monthly BTC hash rate change% in Fig 11. The stability of this smoothed variable in the recent period confirms the rationality of our assumptions of the future BTC network hash rate evolution for the purpose of the mining profitability model presented in the further section.

Network difficulty presented in Fig 12 follows the BTC network hash rate since there is a linear relation between these two factors presented in the formula 1. The difficulty is

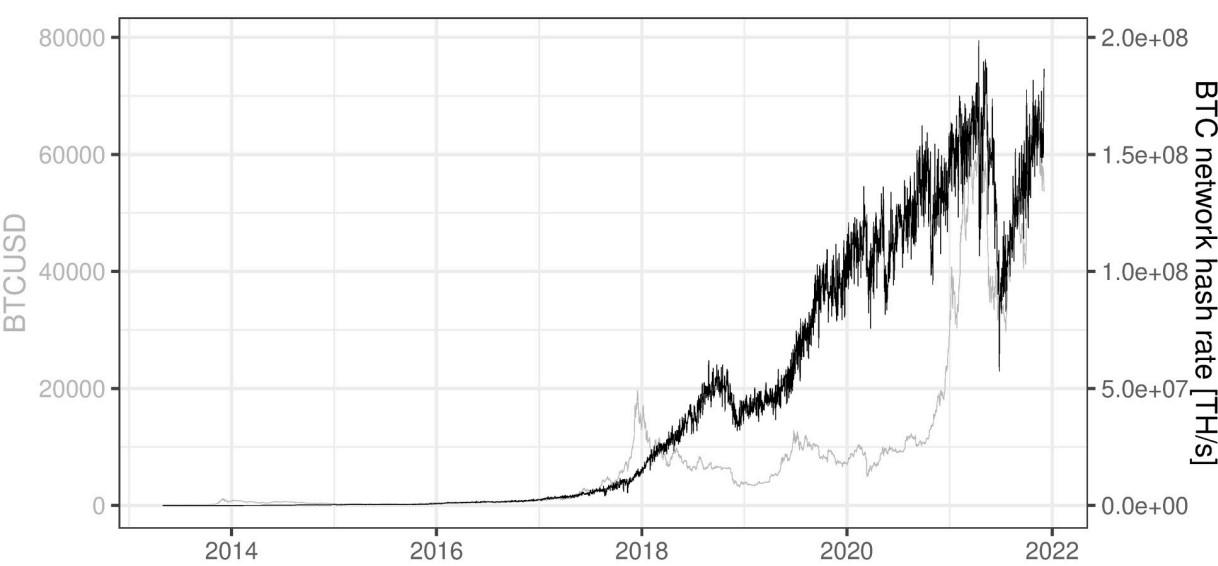

**Fig 10. BTC network hash rate.** Note: Data covers the period between 2013/05/01 to 2021/12/03. Source: https://api.blockchain.info/charts/hash-rate?format=csv.

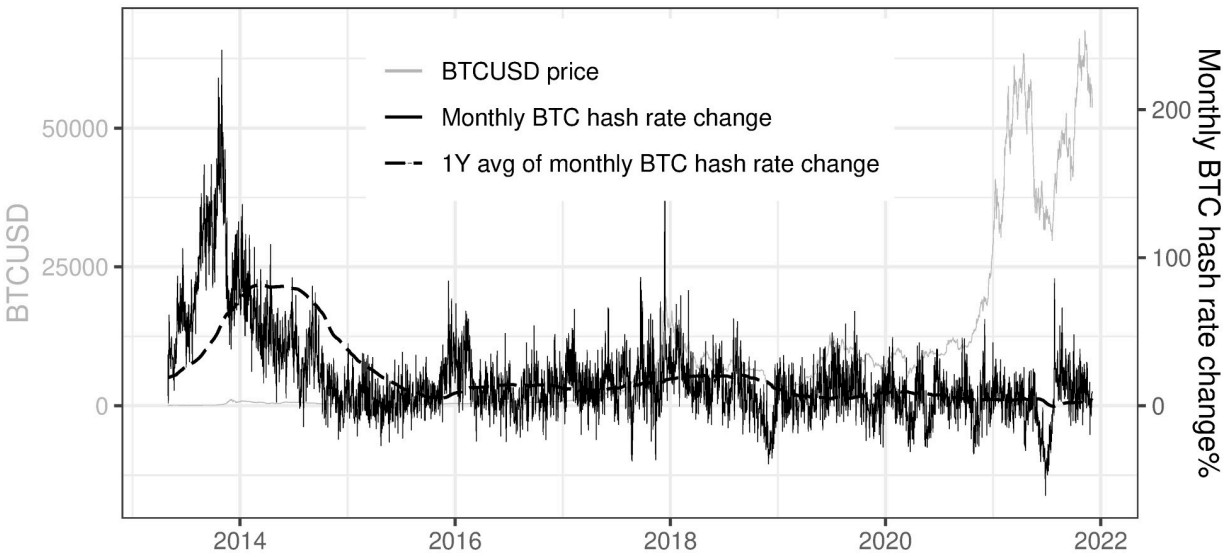

**Fig 11. Monthly BTC hash rate change% and 1Y average of these changes.** Note: Data covers the period between 2013/05/01 to 2021/12/03. Source: https://api.blockchain.info/charts/hash-rate?format=csv.

periodically adjusted (every 2016 block) according to the real hash rate evolution in order to assure predefined intervals between confirmation of consecutive BTC blocks (10 minutes). Figs 10 and 12 can be treated as a depiction of the second research question stated at the beginning (RQ2): *Is there a strong positive relation between BTCUSD price and difficulty/hash rate of BTC network?* We can observe that such positive relations generally exist in the long term (years) but are not always very strong in the short term (months). This conclusion is based on three observations:

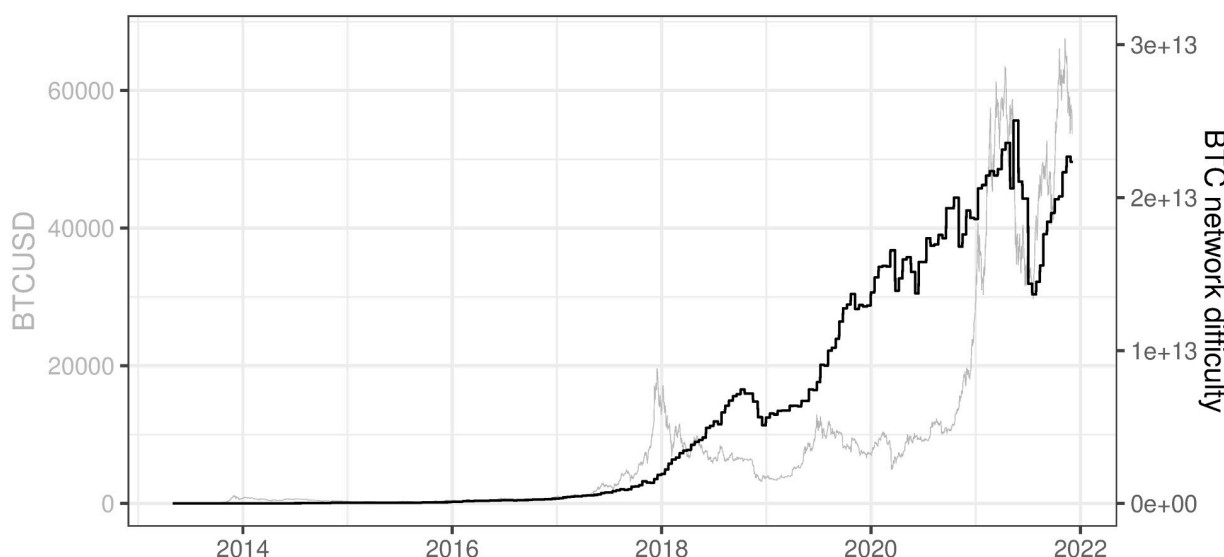

**Fig 12. BTC network difficulty.** Note: Data covers the period between 2013/05/01 to 2021/12/03. Source: https://charts.coinmetrics.io/network-data/.

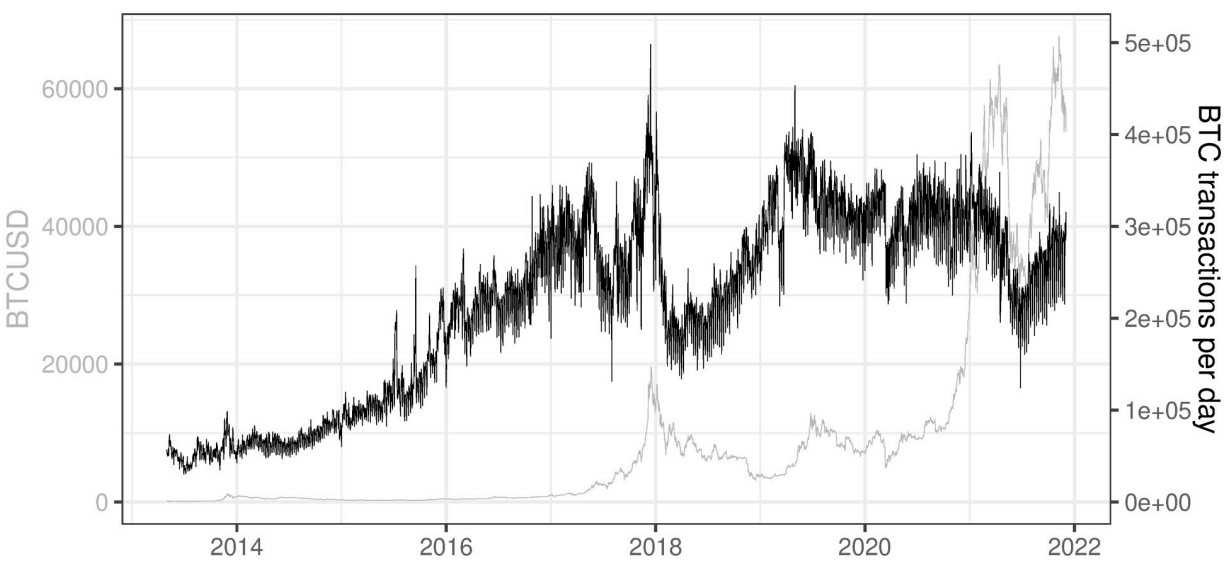

**Fig 13. BTC transactions per day.** Note: Data covers the period between 2013/05/01 to 2021/12/03. Source: https://charts.coinmetrics.io/network-data/.

- There are usually only temporary downward movements in BTC difficulty or hash rate in the recent period, besides the severe one in mid-2021, i.e. we observe very strong upward movements with quite rare corrections, no matter what happens on BTCUSD price,

- A detailed analysis of the very strong downward movements in BTCUSD price (2014 year and then the first half of 2018) and both BTC network difficulty and hash rate shows that difficulty and hash rate grew significantly no matter what happens with BTCUSD price

- The situation was completely different in mid-2021 when the drop and consecutive rise in BTCUSD price and the difficulty and the hash rate of the BTC network happened simultaneously.

A more formal analysis of this research question will be presented in the Main results section (Section 5).

As seen in Fig 13, the number of transactions per day was steadily increasing until mid-2017. After a short period of decline in mid-2017 it reached its high in late 2017 to go down in December 2017 and January 2018. Currently, we can see that the number of transactions stabilized at the level of 3e+05 in the period between 2017 and 2021 year. However, the number of transactions could be biased in the BTC network due to the fact that part of the payments is batched into combined transactions. The batch process is used mainly by industrial players like exchanges, mixers, payments processors, and mining pools (https://coinmetrics.io/batching/). Nevertheless, our profitability analysis is not affected by this fact since we base it on the transaction fee generated on the daily basis rather than on the number of transactions.

The number of daily mined BTC is presented in Fig 14. The values on the chart follow a rule embedded in the mining algorithm saying how many BTC coins are created once the block is confirmed and how often should it happen (approx. every 10 minutes). Since 2016/07/09 the block reward is 12.5 BTC while prior to this date the reward was 25 BTC. Reward halving happens roughly every 4 years (more precisely every 210.000 blocks) and the last one (to

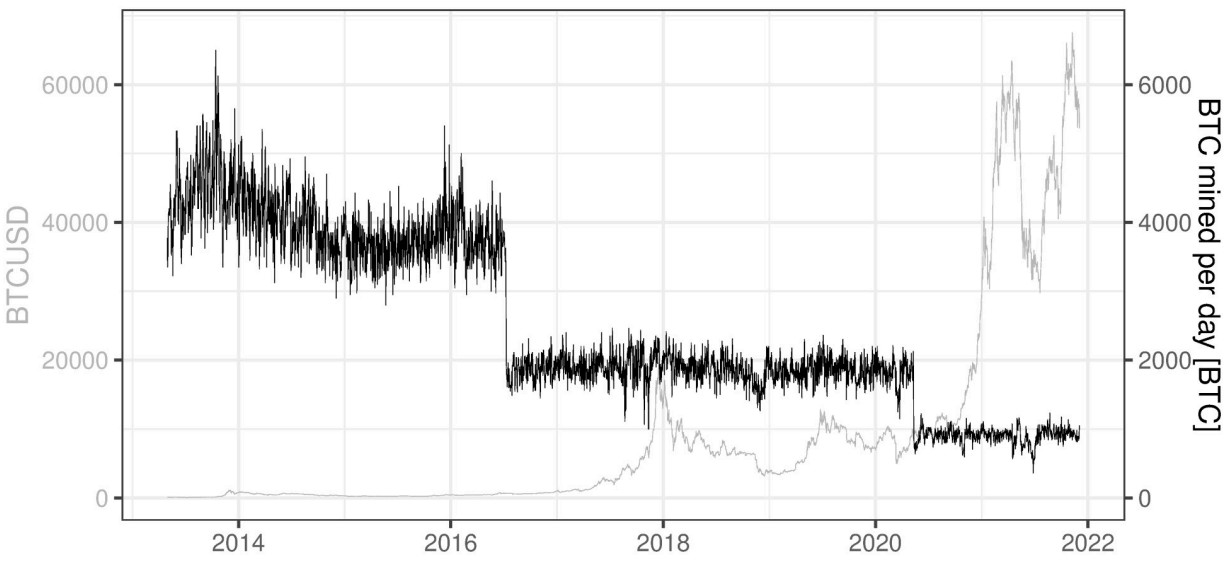

**Fig 14. BTC mined per day.** Note: Data covers the period between 2013/05/01 to 2021/12/03. Source: https://charts.coinmetrics.io/network-data/.

6.25 BTC) was in 2020–05-11 (https://www.bitcoinblockhalf.com/). Current theoretical BTC mined per day should be 900 BTC (=6 · 24 · 6.25 BTC) but in practice it is a slightly higher number (between 900 and 931 BTC for the last year) because BTC network difficulty change is slower (every 2016 blocks) than BTC hash rate change (immediately after plugging in of the new mining machine) (https://coinmetrics.io/on-data-and-certainty/).

Fig 15 shows that the level of BTC transaction fees per day is characterized by high volatility, especially during 2017 and early 2018. The last surge in the transaction fees corresponds to the

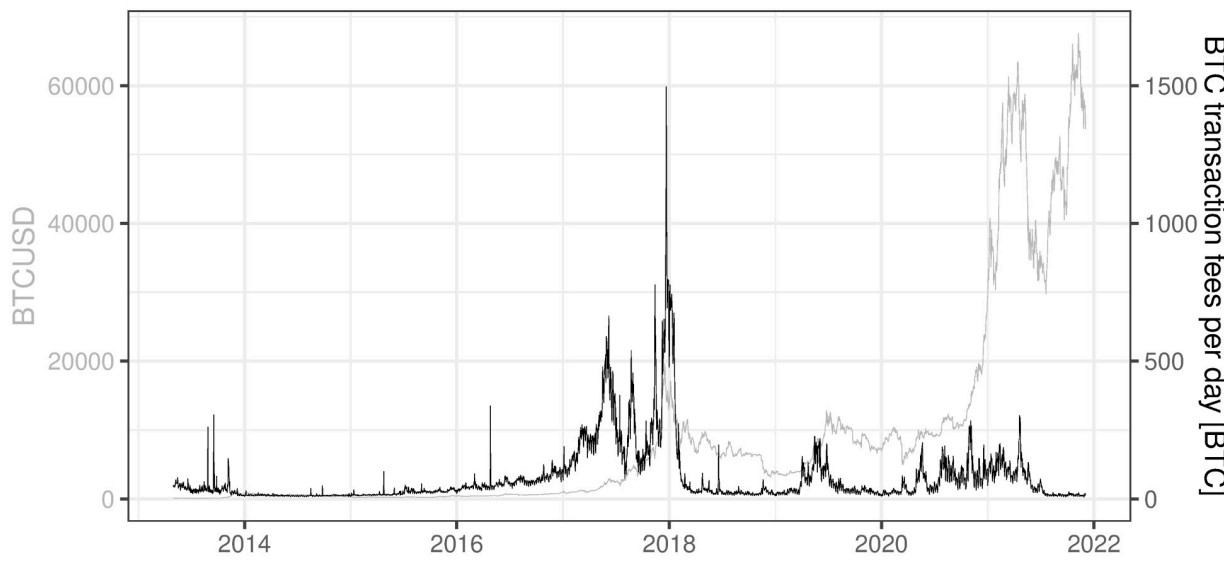

**Fig 15. BTC transaction fees per day.** Note: Data covers the period between 2013/05/01 to 2021/12/03. Source: https://charts.coinmetrics.io/network-data/.

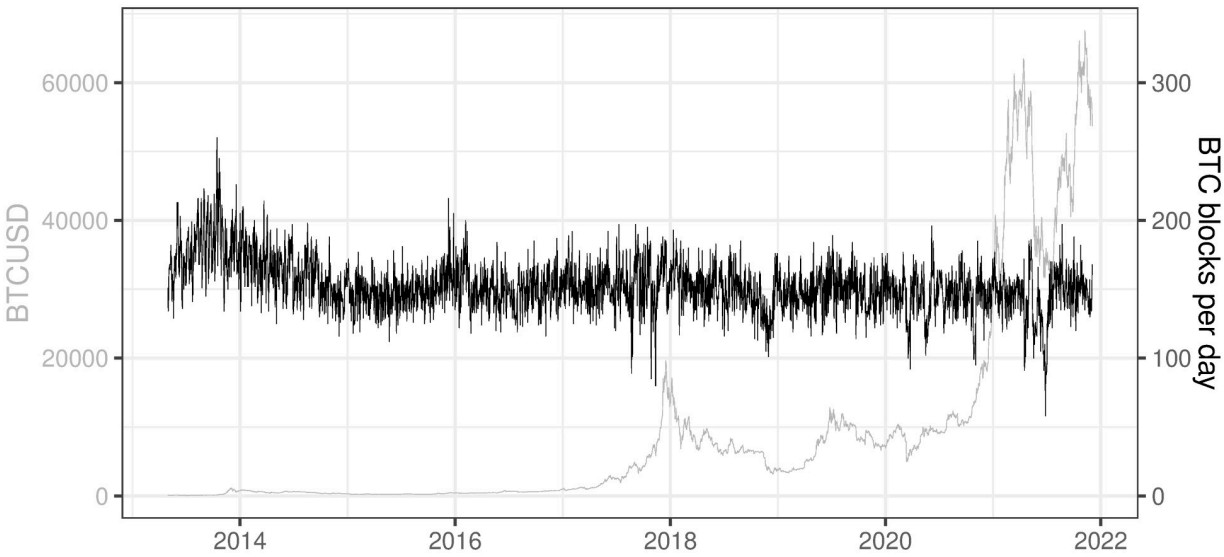

**Fig 16. BTC blocks per day.** Note: Data covers the period between 2013/05/01 to 2021/12/03. Source: https://charts.coinmetrics.io/network-data/.

soaring prices and huge volatility of BTC and then a steep decline in its price. The most recent values remain, for the last several months, at a relatively low level of 65 BTC in transactions fees on average per day for the last year and 14 BTC in transactions fees on average per day for the last quarter.

The number of blocks confirmed every day is determined by the mining algorithm which says that, on average, a block should be confirmed every 10 minutes resulting in a theoretical value of ca. 144 blocks per day ( = 6 · 24). This interval may diverge from the target in the short period (see Fig 16) hence the network difficulty adjustment is performed to assure the convergence to the target in the longer term period. This phenomenon is explained in more detail in the paragraph describing Fig 14.

For the purpose of mining profitability calculation, we decided to use BTC transaction fees presented as a % of mined BTC coming from block reward for the confirmation process (according to formula 5, referred further as Fees%). As one can observe in Fig 17 Fees% are highly volatile. The one-year average of this ratio (referred further as 1Y avg of Fees%) is presented in Fig 17 as well. The current level of the average was taken as a current base case scenario for further analysis of BTC mining profitability. This is rather a conservative approach and may lead to a slight overestimation of the mining profitability when compared to the actual market state, but we are aware of that fact.

The charts presented above gave us the ability to rethink the most appropriate methodology for the purpose of this study and set the adequate values of our variables for the base case scenario.

## 4 Methodology

### 4.1 Model description

BTC mining profitability is calculated based on the following equations:

$$\text{BTC mining profitability} = \text{PnL\%} = \frac{\text{Amount earned} - \text{Amount invested}}{\text{Amount invested}}, \qquad (6)$$

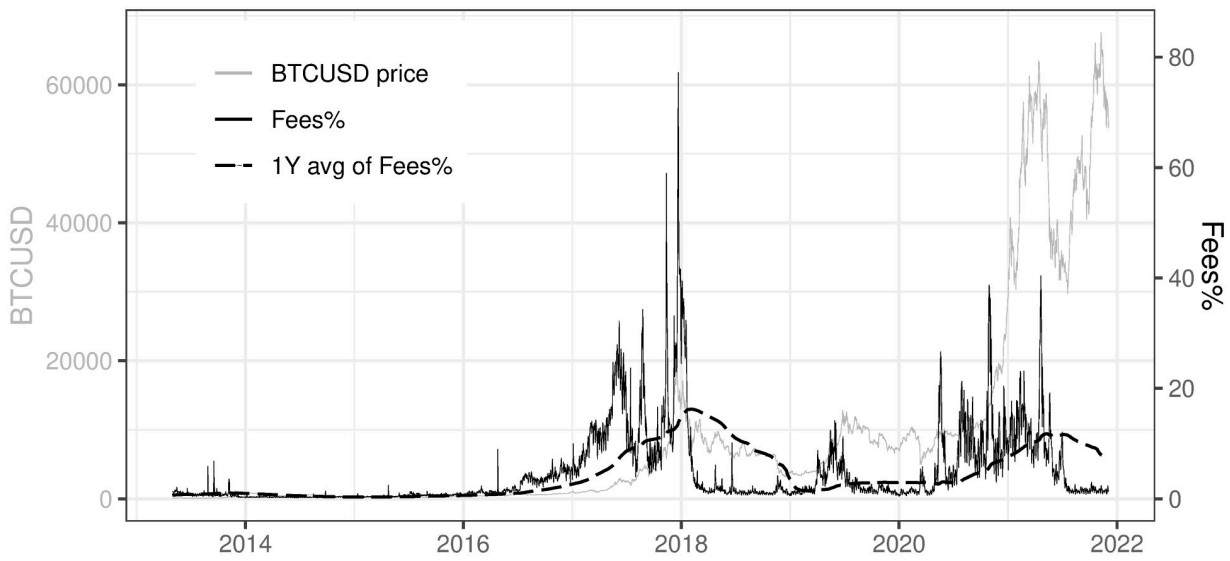

**Fig 17. Fees% and 1Y average of Fees%.** Note: Data covers the period between 2013/05/01 to 2021/12/03. Source: https://charts.coinmetrics.io/network-data/.

where:

$$
\begin{aligned}
\text{Amount earned [USD]} = \quad & \text{Value of BTC mined per year [USD]} \\
& + \text{Transaction fees received per year [USD]} \quad (7) \\
& + \text{Residual value of hardware after one year [USD]},
\end{aligned}
$$

$$
\begin{aligned}
\text{Amount invested [USD]} = \quad & \text{Energy cost per year [USD]} + \text{Hardware cost [USD]} \\
& + \text{Maintenance cost per year [USD]}, \quad (8)
\end{aligned}
$$

$$
\begin{aligned}
\text{Value of BTC mined per year [USD]} = \quad & \text{BTCUSD} \cdot \frac{\text{Theoretical BTC mined per year}}{12} \\
& \cdot \frac{\text{Hardware hash rate}}{\text{BTC network hash rate}} \cdot \frac{1 - \left(\frac{1}{1+d}\right)^{12}}{1 - \frac{1}{1+d}}, \quad (9)
\end{aligned}
$$

and

$$
d = 1\text{Y avg of monthly BTC hash rate change\%}, \quad (10)
$$

$$
\begin{aligned}
\text{Theoretical BTC mined per year} = \quad & \text{Theoretical BTC blocks mined per hour} \\
& \cdot \text{BTC block reward} \cdot \text{Days per year} \cdot \text{Hours per day} \quad (11)
\end{aligned}
$$

$$\text{Transaction fees received per year [USD]} = 1\text{Y avg} \quad \text{of Fees\%}$$
$$\cdot \text{Value of BTC mined per year [USD],} \tag{12}$$

$$\text{Residual value of hardware after one year [USD]} = \% \text{ of new} \quad \text{Hardware cost}$$
$$\cdot \text{Hardware cost [USD],} \tag{13}$$

$$\text{Hardware net price [USD]} = \text{Mining machine net price [USD]} + \text{PSU net price [USD],} \tag{14}$$

$$\% \text{ of new hardware cost} = \frac{\text{estimated Hardware cost after } 12 \text{ months}}{\text{current Hardware cost}} = \left(\frac{1}{1+d}\right)^{12}, \tag{15}$$

$$\text{Energy cost per year} = \frac{\text{Hardware power consumption [W]}}{1000} \cdot \text{Unit energy price [USD/kWh]}$$
$$\cdot \text{Days per year} \cdot \text{Hours per day,} \tag{16}$$

$$\text{Hardware cost [USD]} = \text{Hardware net price [USD]} \cdot (1 + \text{Taxes\&Shipment\%}), \tag{17}$$

$$\text{Maintenance cost per year [USD]} = \text{Maintenance ratio\%} \cdot \text{Value of BTC mined per year [USD].} \tag{18}$$

### 4.2 Model assumptions. Base case scenario

- Assumption 1. BTCUSD price in *t* months is the same as today.

- Assumption 2. USD energy prices during the next *t* months are the same as today. For the base case scenario, we assume China's unit energy price but in sensitivity analysis, we refer to other energy prices as well.

- Assumption 3. Monthly changes in the BTC network hash rate are approximated by their 1Y average.

- Assumption 4. The constant amount of BTC mined monthly by the entire BTC network (specifically, the halving of BTC block reward not assumed during the time considered).

- Assumption 5. Maintenance cost is a fraction of the value of BTC mined per year [USD].

- Assumption 6. The residual value of our mining machine is based on the percentage decrease of monthly BTC mined in each consecutive month. In order to evaluate "% of new Hardware cost" we assumed that Hardware price will diminish according to its shrinking participation in increasing BTC network hash rate, which can be estimated mainly by factor d (1Y avg of monthly BTC hash rate change%) from formula 10.

- Assumption 7. We assume that the amortization period for our hardware is 12 months.

- Assumption 8. Changes in Fees% are approximated by their 1Y average.

**Table 3. Monthly USD profitability of mining of single Antminer S19J Pro (104 TH/s) with adjustment of difficulty based on 1Y avg monthly BTC hash rate change.**

| month | mining of single Antminer S19J Pro with adj. of difficulty | USD value of mined BTC | USD mining fees | USD hardware cost | USD energy cost | USD maintenance costs | USD profit before hardware | USD profit after hardware |
|---|---|---|---|---|---|---|---|---|
| 1M | 0.015 | 819.177 | 61.166 | 1252.96 | 201.568 | 204.794 | 473.981 | -778.979 |
| 2M | 0.015 | 790.399 | 59.017 | 1252.96 | 201.568 | 197.600 | 450.249 | -802.711 |
| 3M | 0.014 | 762.632 | 56.943 | 1252.96 | 201.568 | 190.658 | 427.350 | -825.610 |
| 4M | 0.014 | 735.841 | 54.943 | 1252.96 | 201.568 | 183.960 | 405.256 | -847.704 |
| 5M | 0.013 | 709.990 | 53.013 | 1252.96 | 201.568 | 177.498 | 383.938 | -869.022 |
| 6M | 0.013 | 685.048 | 51.150 | 1252.96 | 201.568 | 171.262 | 363.369 | -889.591 |
| 7M | 0.012 | 660.982 | 49.354 | 1252.96 | 201.568 | 165.246 | 343.523 | -909.437 |
| 8M | 0.012 | 637.762 | 47.620 | 1252.96 | 201.568 | 159.440 | 324.373 | -928.587 |
| 9M | 0.011 | 615.357 | 45.947 | 1252.96 | 201.568 | 153.839 | 305.897 | -947.063 |
| 10M | 0.011 | 593.739 | 44.333 | 1252.96 | 201.568 | 148.435 | 288.069 | -964.891 |
| 11M | 0.011 | 572.881 | 42.775 | 1252.96 | 201.568 | 143.220 | 270.868 | -982.092 |
| 12M | 0.010 | 552.755 | 41.273 | 1252.96 | 201.568 | 138.189 | 254.271 | -998.689 |
| sum | 0.152 | 8136.565 | 607.533 | 15035.52 | 2418.811 | 2034.141 | 4291.145 | -10744.375 |

Note: The analysis assumes current mining conditions as of 2021/12/03 summarized in Table 2.

### 4.3 Values of variables as of 2021/12/03

All our assumptions for BTC mining profitability purposes can be found in Table 2. The base case scenario in Section 5 will be presented based on this assumption.

## 5 Main results

In this section, we wanted to tackle the base case scenario in which we consider our investment before the hardware is bought (Section 5.1) and the second variant in which we analyze BTC mining investment after all the necessary hardware was bought (Section 5.2). All analyses presented in this paper are based on the same assumptions as in Section 5.1. Nonetheless, Section 5.2 was added in order to refer to the situation of an investor who is already engaged in BTC mining activity and already bought the mining machine.

### 5.1 Base case scenario (the consideration before the hardware is bought)

This scenario assumes that we are before the decision on whether to start mining operations and we try to consider all possible revenues and costs under current market conditions and make the correct decisions based on mining profitability.

Our main results show that BTC mining profitability is on the level of -4.9%, which means that during 12 months we would incur -0.049m USD of loss on each 1m USD invested in BTC mining. The details of our calculations, month by month, are presented in Table 3 and Fig 18. It means that referring to our main hypothesis (RH1) we can say that BTC mining is not profitable under current market conditions existing at the beginning of December 2021.

What is more important, it could affect the state of the BTC blockchain environment as a whole, taking into account that mining activity, as a necessary condition for block confirmation, is an essential part oftheBTC network. At this point, we come to the fifth research question (RQ5): *What are the consequences of BTC mining efficiency for the future ability of blocks to be mined further?* and we would like to describe some scenarios.

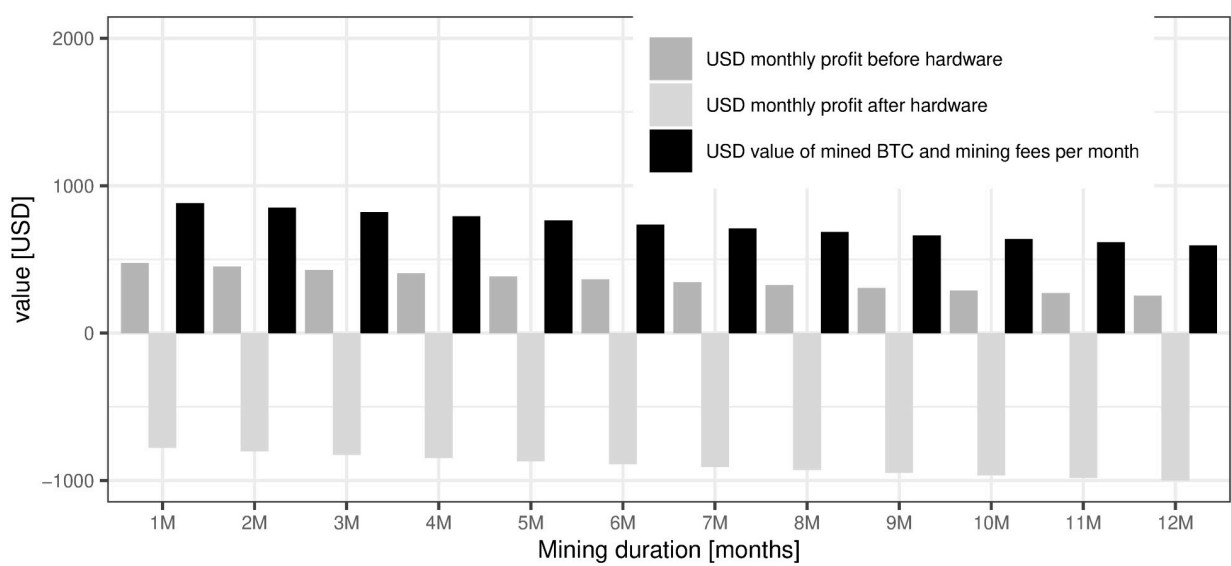

**Fig 18. The profitability of mining in USD.** Note: The analysis assumes current mining conditions as of 2021/12/03.

To start with the fundamental economic assumption, it is important to notice that the current state, when mining is not profitable, is not sustainable in a long term. Periodical divergence from the equilibrium state, defined as a state negligibly close to the breakeven point, may appear especially when the industry is not mature enough. However, the BTC ecosystem has to revert to equilibrium if it is going to survive. We present six theoretical scenarios being an attempt to refer to RQ5, nevertheless one may imagine a mix of these scenarios pushing BTC mining toward the breakeven point. In scenarios from one to five, we assume that the BTC network will still exist and be able to evolve, while in the last one, we take into account that current mining unprofitability could weaken and destroy the network as a whole, what in reality has very low probability at this time.

- The first scenario is connected with a sharp increase in BTCUSD triggered by current mining unprofitability. However, it may create a simultaneous sociological effect attracting more miners to the network, thus postponing equilibrium further toward higher BTCUSD levels. The miners may decide to undertake currently not profitable mining in a hope of extraordinary BTCUSD increases would make their activity profitable. Assuming that miners behave like "homo oeconomicus" they should turn towards investing in BTCUSD derivatives instead of mining activity if they believe in substantial growth of BTCUSD.

- The second scenario is a substantial increase in the transaction fee level. This is a likely scenario after the reward halving which occurred in 2020, and also a case considered by theoreticians and practitioners.

- The third theoretical scenario is that a number of miners simply switch off their machines after suffering some losses due to persistent negative profitability (even not enough to cover operational expenditures—OPEX), thus BTC network hash rate eventually drops making mining profitable again. From this point, we see two reasonable paths of how the situation develops further in this scenario. The first one assumes that miners may keep the hardware switched off during the period of negative profitability in the hope to switch them on when mining is profitable again. Alternatively, miners may sell the machines and invest the

proceeds into BTCUSD derivatives if they still want to keep positive exposure to BTCUSD price. Currently, this scenario has a low probability of realization because as we could see in the ninth column of Table 3 the mining is profitable if we assume that an adequate mining machine is already bought.

- The fourth scenario assumes that a major breakthrough in the hardware happens, e.g. a new generation of ASICs is introduced into the market. This scenario may change the BTC mining landscape in the same way as CPU/GPU and GPU/ASIC migrations had before. Since the energy and hardware prices vary independently, they are not able to bring the industry to the breakeven point. Therefore, they should coexist with other drivers in a combined scenario.

- The fifth scenario covers a change in the block confirmation method connected with an enormous limitation of energy consumption, i.e. the switch from Proof-of-Work to Proof-of-Stake, which could significantly change the main important conclusions of this analysis.

- The sixth scenario is a catastrophic one for the BTC network. It deals with the possibility that the mining profitability problems and any potential issues, e.g. with block confirmation, undermine the credibility of the whole BTC blockchain and may be a cause of its downfall. We humbly assume this scenario is currently not the most probable one, but nevertheless, we have to take it into account.

### 5.2 Base case scenario BIS (the consideration after the hardware is bought)

Before we delve further into the sensitivity analysis of our results we would like to show how long should we mine BTC within our 12 months amortization period taking into account only operational costs (energy and maintenance)—under the assumption that our mining machine has already been bought. This analysis assumes that our mining machine was bought in the past and currently we can make the decision to mine or not to mine considering operational expenses (OPEX) and excluding hardware costs (CAPEX). Table 3 shows the numbers for such analysis.

Table 3 and especially Fig 18 clearly shows that, if we do not take into account any hardware cost, the profitability of mining looks different. For the whole amortization period(12 months) we are able to cover expenses for energy and maintenance costs even though each month the number of mined BTC diminishes due to the increase of the BTC network hash rate. This is the confirmation of the low probability of the third scenario from the previous subsection.

This is a very good place to refer to the fourth research question (Q4): *What is the rational level of the amortization period for our mining machines?* Based on the analysis of Table 3 and Fig 18 we can say that this period can be longer than 12 months, i.e. until the moment our mining machine is useless, meaning that we will not even be able to cover OPEX costs. In the real life, if we take into account the monthly amortization costs of our mining machine, we will realize that the profit even for the first month is negative. The above-mentioned consideration clearly confirms our initial calculations of mining profitability presented in subsection 5.1 of this section. Therefore, we reach the conclusion that one definitely should not start mining under the current market conditions with these assumptions. Hence, a rational level for the value of the amortization period does not exist in this specific case (We have to take in mind that in order to present any calculations concerning profitability we have to assume some level of the amortization period. We decided to set it to 12 months but our results clearly show that the length of the amortization period is strictly negatively correlated with %

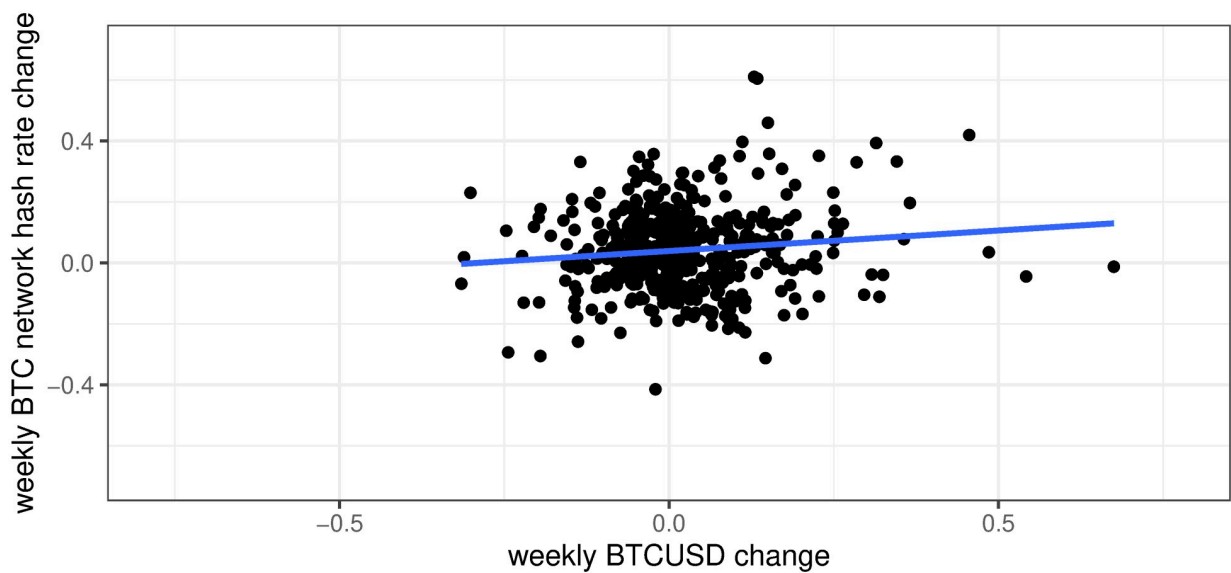

**Fig 19. Weekly BTCUSD change vs weekly BTC network hash rate change.** Descriptive statistics for BTCUSD weekly change: n = 448, min = -0.315, max = 0.675, mean = 0.02, median = 0.003, std dev = 0.116, skewness = 1.063, kurtosis = 3.875. Descriptive statistics for hashrate weekly change: n = 448, min = -0.414, max = 0.611, mean = 0.042, median = 0.035, std dev = 0.137, skewness = 0.438, kurtosis = 1.113. Note: Data covers the period between 2013/05/01 to 2021/12/03. Own calculation based on the data from: https://charts.coinmetrics.io/network-data/, https://api.blockchain.info/charts/hash-rate.

profitability of mining and that its optimal level should be set to zero, i.e. no mining activity should be undertaken, as is shown on Fig 25.) (see Fig 25).

One can definitely argue that the horizontal trend assumption for future BTCUSD price or our assumptions concerning the monthly change of BTC network hash rate are too rigorous or even arbitrary. Therefore, to address such a potential discussion, we present a very detailed and rigid description of these assumptions in sections 4 and 5 referring to the current market conditions in order to avoid possible allegations concerning these issues.

## 5.3 Other research questions

i. RQ1: What is the sensitivity of BTC mining profitability to initial assumptions and future trajectories of main parameters? → we address this question in the Sensitivity analysis section.

ii. RQ2: Is there a strong positive relation between BTCUSD price and the difficulty/hash rate of the BTC network?

This question was already partially addressed in the third section. Nevertheless, below we try to present a more formal analysis based on the time series of BTCUSD and BTC network difficulty/hash rate to gauge a potential relation between BTCUSD price and the difficulty/hash rate of the BTC network. As both variables are non-stationary, we take 7 days' returns in the historical values which allows us to calculate Pearson's correlation coefficient. The results confirm low relation between BTCUSD price and the hash rate of BTC network, which additionally increased significantly in the last year. Pearson's coefficients are at the level of 0.11, 0.27, 0.29 for the periods: 2013/05/01–2021/12/03, last 12 months, and the whole 2021, respectively. Moreover, a low correlation can be seen in Fig 19 which presents weekly BTCUSD change vs

weekly BTC network hash rate change. Therefore, we can confirm that there is a positive relation between BTCUSD price and the difficulty/hash rate of the BTC network but this relation was low in the past while currently, it seems to be much stronger.

iii. RQ3: What are the breakeven levels of the main BTC mining parameters in order to make this activity profitable? → we refer to this question in detail in the Breakeven section (Section 7).

iv. RQ4. What is the rational level of the amortization period for our mining machines? → we refer to this question in detail in the Main results part in subsection 5.2.

v. RQ5: What are the consequences of BTC mining efficiency for the future ability of blocks to be mined further? → we refer to this question in detail in the Main results part in subsection 5.1.

vi. RQ6: Can we expect that blocks will not be confirmed if the efficiency of BTC mining is negative for an extended period of time?

This is one of the most important questions in this paper because this issue affects not only single mining endeavors but can influence the probability of the BTC network working properly as a whole. The answer is NO, at least for a longer period of time. Blocks will be mined unless there is trust in the BTC blockchain protocol, regardless of BTCUSD price and other variables. BTC ecosystem has a feature to adjust itself accordingly to the observed parameters. Let us rephrase the situation elaborated in a possible scenarios evolution. The long period of negative mining profitability motivates part of the miners to disconnect from the network thus pushing the network hash rate down until the mining is profitable again. In such a situation there is a special case when blocks can be mined for a much longer period of time especially if more miners would decide to stop mining at the same time. Nevertheless, it will last only until the next difficulty adjustment, so not longer than until the next 2016 blocks will be mined.

Another potential consequence is that a lower hash rate may make it easier for entities with plenty of spare mining capacity to attack the network. However, we believe that BTC network at its current state of development limits substantially the probability of such an event.

vii. RQ7: Should we invest in BTC mining if we expect the BTC price to increase substantially?

The answer is quite straightforward: definitely NO. If one expects the BTCUSD price to increase, they should rather invest directly in BTC spot or through BTC derivatives in order to capitalize on their predictions. Here we can provide a very simple analogy of BTC mining and GOLD mining. Should we really invest in gold mining embracing all potential equipment and operational issues if we think that the price of gold will go up? Probably we would have higher operating leverage from such investment in comparison to direct investment in gold, but all the risks connected with setting up such activity would affect heavily our final profitability.

viii. RQ8. Is it possible that technological advancements in ASICs performance would change the BTC mining landscape and make mining profitable?

All kinds of technical advancements, especially those focused on improving hardware performance, could change the landscape of the BTC mining industry. Breakthroughs in the efficiency, e.g. a new generation of ASICs, could make a difference, but the scale and direction of the change will definitely depend on the market price of new hardware. Should a new machine boost the profitability of mining, then its price should be economically justified by the business models in the mining industry. Another thing is that technological improvements would

probably once again push the BTC network hash rate upward again and in spite of the much higher efficiency of a single machine, nothing would change for the network as a whole.

ix. RQ9: What kind of new financial instruments may stimulate BTC mining industry?

Analyzing the cost and revenue sources in the mining industry, we may conclude that part of the cost factors has its own, relatively well-developed, derivatives market for hedging purposes. BTCUSD can be fairly easily hedged via futures contracts or options strategies. Energy prices have tradable commodity futures and forwards, even though we assume that the biggest miners have individual contracts with energy plants. The hardware and maintenance costs are exogenous (except for cases when big producers have their own mining subsidiaries, e.g. Bitmain), but not quite tradable. However, there is one important and also an exogenous driver of profitability—the hash rate. This part of the business case is nowadays not hegdeable. BTC hash rate may be treated as an index (Aware of the technical problems that might arise in the construction of such an index and derivatives based on it (due to its mainly growing nature) we advise that a basis instrument should be constructed as a departure from a projected long-term change of BTC network hash rate (1m, 3m, 6m, 1y, etc.). The introduction of derivatives on such an index would enable one to either speculate or hedge against adverse scenarios.) and even be considered as an asset. Furthermore, an introduction of futures and options market for this commodity would without any doubt cause a tremendous stimulation of the industry creating an opportunity for further development of the BTC network, along with an ability to stabilize the performance and hedge mining industry against adverse scenarios.

## 6 Sensitivity analysis

In this section, we present a detailed analysis of mining profitability assuming different trajectories of cost and revenue-driving parameters. This analysis is presented in order to address the first research question (RQ1): *What is the sensitivity of BTC mining to initial assumptions and future trajectories of main parameters?* The simulations are performed with ceteris paribus assumption i.e. changing only one variable at a time, while all other parameters remain constant.

One may raise an objection that such an approach is far from the real market dynamics. Nevertheless, we stick to this way of presentation as we see it as the most optimal description of the market's possible states. We present the possible market evolution while simultaneously keeping the analysis at a fairly simple level for the business interpretation and conclusion sake. One- and two-factor sensitivity analysis allows us to present our results in a form of conclusive charts. A multi-factor simulation process has been also created and the results will be published in the next papers.

### 6.1 One-factor analysis ceteris paribus all others

i. BTCUSD price

Fig 20 presents an obvious positive relation between the profitability of mining and the quotation of BTCUSD. Under current market conditions, one may conclude that in order to reach breakeven the BTCUSD price should increase to 61313. One can easily make a conclusion on marginal profitability change for the defined price of BTC. The slope of the line characterizes current market conditions for other variables. The change in the assumed level of other variables will move the position and slope of this line. These effects are presented further in this section.

ii. Monthly hash rate increase

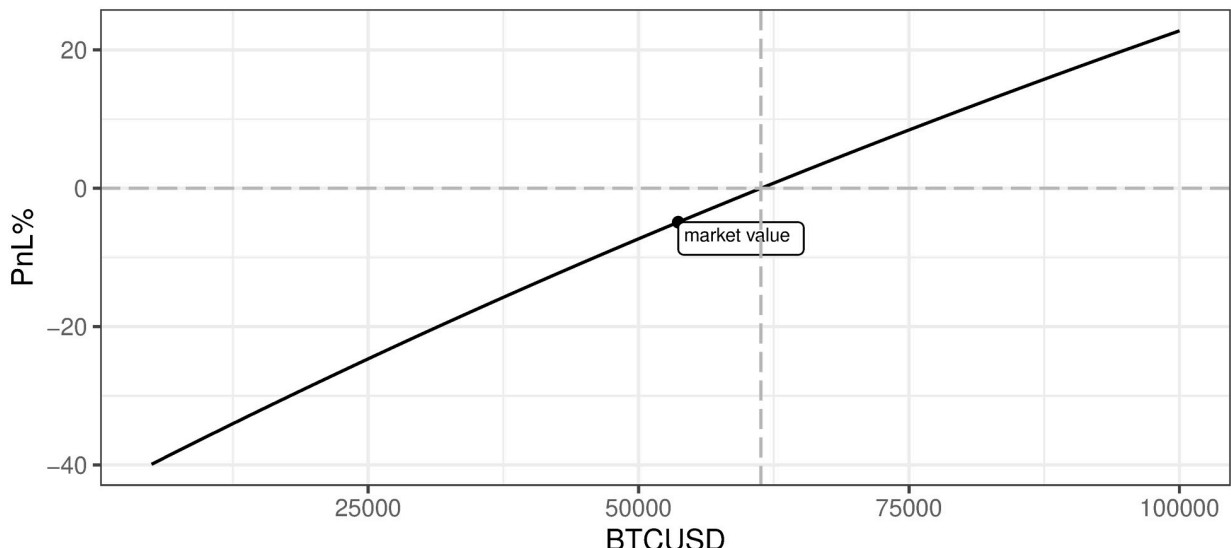

**Fig 20. BTC mining profitability (PnL%) for different BTCUSD prices.** Note: PnL%, calculated according to the formula 6. BTCUSD hypothetical prices range from 5000 up to 1e+05 USD. The market value point describes the current state of market mining efficiency assuming the parameter outlined in the section Values of variables as of 2021/12/03. Source: own calculations based on data described in the section Crucial assumptions and data.

Fig 21 presents a negative relation between the dynamics of the monthly BTC network hash rate change and the mining profitability. This has to be a crucial part of the business case model when analyzing the mining activity start-ups. Two main conclusions from Fig 21 are: (a) assuming that the monthly BTC network hash rate change is on the level of the last year's

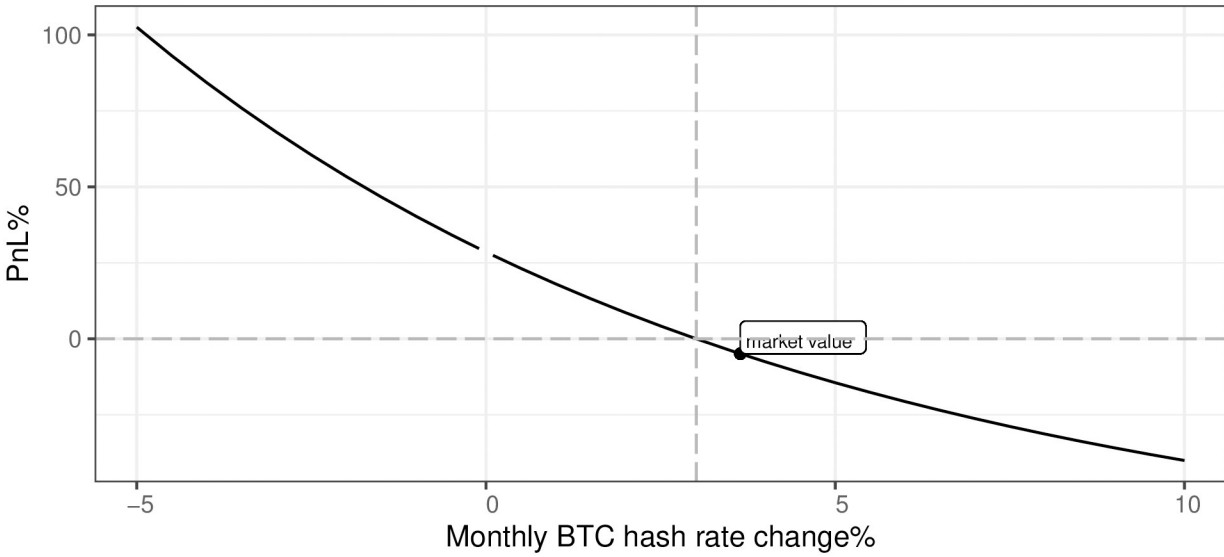

**Fig 21. BTC mining profitability (PnL%) for different evolution of network hash rate.** Note: PnL%, calculated according to the formula 6. Hypothetical monthly BTC hash rate change ranges from -5% up to 10%. Negative values mean the BTC network hash rate decreases, while positive describe an increasing BTC network hash rate. The market value point describes the current state of the market mining efficiency assuming the parameter outlined in the section Values of variables as of 2021/12/03. Source: own calculations based on data described in section Crucial assumptions and data.

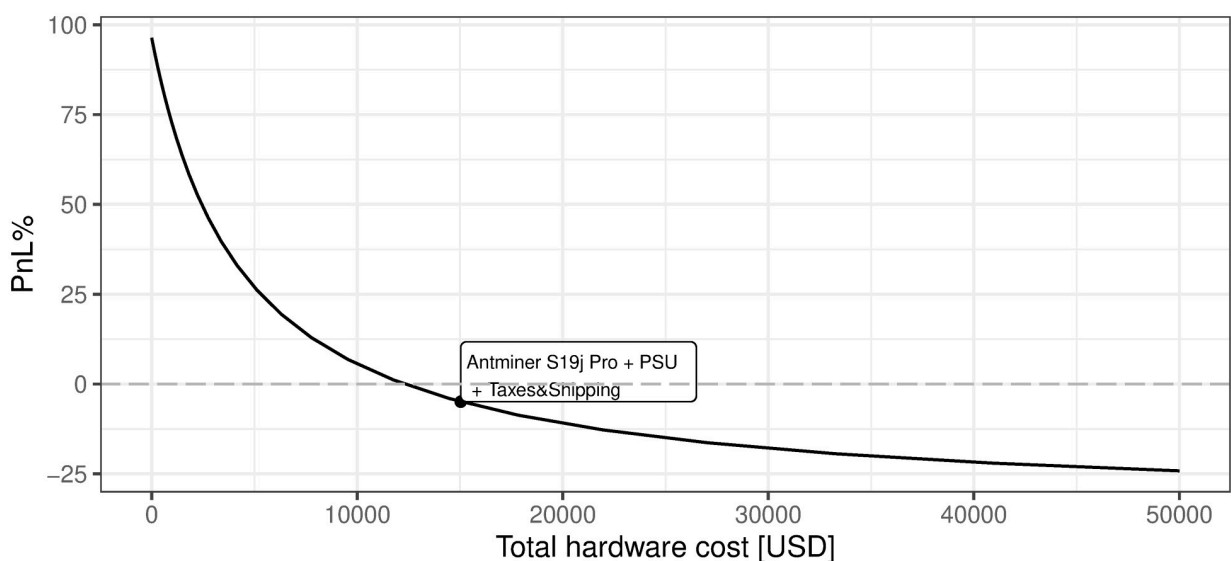

**Fig 22. BTC mining profitability (PnL%) for different cost of hardware.** Note: PnL%, calculated according to the formula 6. Hypothetical hardware cost ranges from 0 up to 50000 USD per unit. Since the Antminer S19J Pro is defined as the most efficient for mining purposes (in terms of price vs. power consumption and hash rate), its cost is analyzed on the charts. The market value point describes the current state of market mining efficiency assuming the parameter outlined in the section Values of variables as of 2021/12/03. Source: own calculations based on data described in section Crucial assumptions and data.

average, the mining profitability is negative, (b) the sensitivity of the business model to the monthly BTC network hash rate change rises once this variable is lower and profitability surges even faster once the network is shrinking. Under current conditions, one may conclude that monthly BTC network hash rate change can not be higher than 3% if we want to stay with the profitability of mining in the positive territory.

iii. Hardware total cost

As presented in Fig 22, BTC mining efficiency is negatively linked with hardware costs which is an obvious conclusion, but one may also notice that this relation is not linear. The sensitivity increases with the decrease in hardware costs. The current market state refers to the cost of hardware unit at the level of $1.503552 \times 10^4$ USD. The most interesting conclusion is that BTC mining breakeven with regard to hardware costs is currently at the level of 12368 USD. As described in the previous sections—the 12 months of mining revenues currently cover OPEX expenses i.e. electricity, maintenance, and other variable costs, provided current market conditions.

iv. Price of energy per kWh in USD

Fig 23 describes how mining efficiency depends on the energy cost. Industrial energy prices are assumed for the sake of clarity. However, as described in the previous section, the supply of power for the BTC mining industry comes mainly from renewable energy with the dominance of hydro-generated power. Analyzing Fig 23 one can easily conclude what would have been the mining profitability if the energy cost had reached the defined level. The main conclusion is that for the cost of energy lower than 0.054 USD per kWh, the mining activity starting to be profitable. Therefore, if the mining endeavor is able to have the minimum energy cost of 0.01 USD per kWh then such activity for sure should be undertaken.

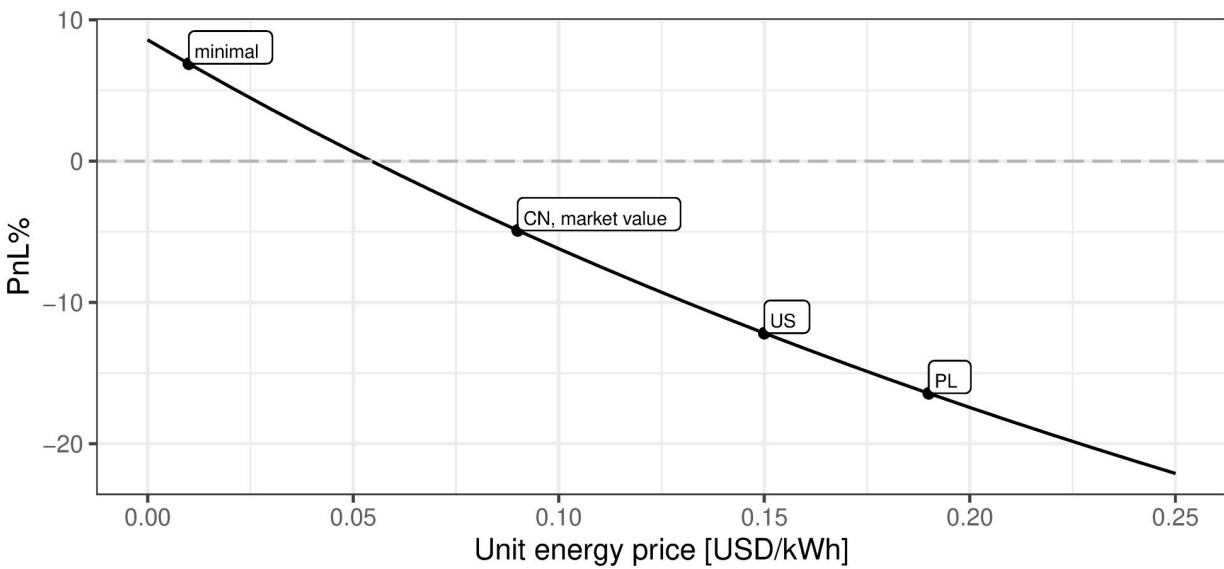

**Fig 23. BTC mining profitability (PnL%) for different prices of electricity.** Note: PnL%, calculated according to the formula 6. Hypothetical electricity prices range from 0 to 0.25 USD per kWh. Minimal, CN, PL, and US define respectively minimal world energy level, China, Poland, and US located mining profitability corresponding to the local energy cost, provided that remaining parameters are equal. The market value point describes the current state of market mining efficiency assuming the parameter outlined in the section Values of variables as of 2021/12/03, where energy cost is assumed to be 0.09 USD per kWh. Source: own calculations based on data described in section Crucial assumptions and data.

v. Transaction fee as a percentage of mining reward (Fees%)

Fig 24 presents the impact of the fees changes on the mining profitability. One can clearly see that the current contribution of fees to the revenues of the activity is very limited. Nevertheless, this part of the business is likely to change over time due to two main reasons: (a) after the block reward was halved in 2020 and there should be some substitute of reward revenue for miners, (b) current mining does not generate sustainable profit and one of the ways to assure the positive business output is to increase the mining costs, which at the same time are revenues for miners. As one can see in Figs 15 and 17 in the previous sections, historical periods with much higher contributions of fee revenue were observed. However, even if we assume historical highs we currently will not be able to reach the breakeven point. Similar research describing this issue in detail can be found in Chepurnoy, Kharin, and Meshkov (2018).

vi. Amortization period

Fig 25 presents the relation between profitability and amortization period. The relation is monotonically decreasing which is a result of the fact that under current market conditions, mining is not profitable from the very beginning. The longer we keep the machine working, the more of the invested capital we lose, mainly on OPEX. The story alters for higher levels of BTCUSD which are presented in the further part of this section.

The conclusions from the factor sensitivity analysis are summarized in Table 4.

## 6.2 Two factors ceteris paribus all others

i. Price of energy per kWh jointly with BTCUSD price

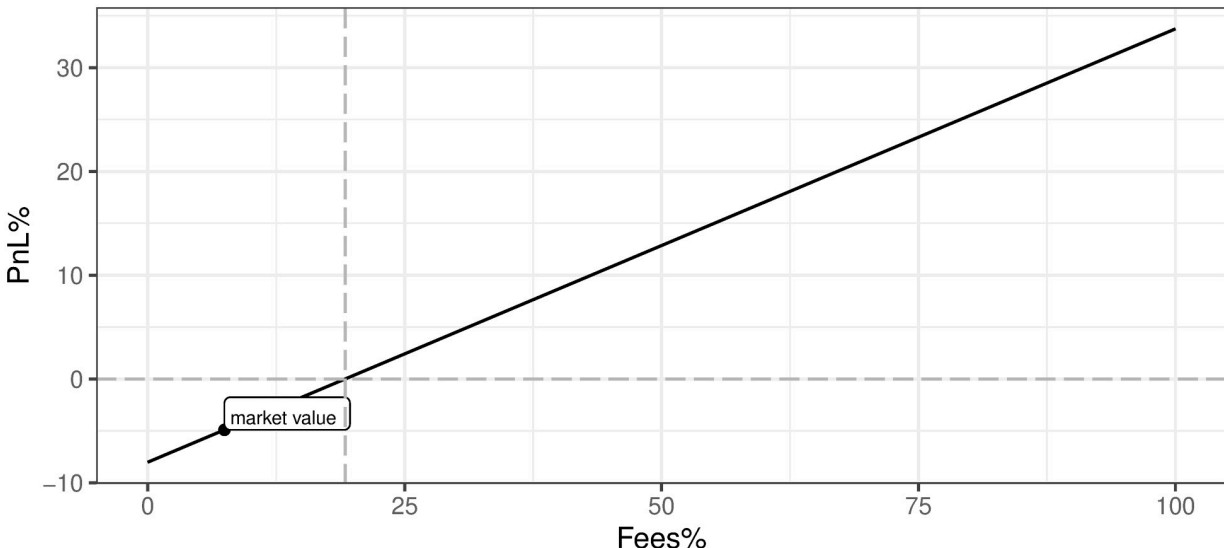

**Fig 24. BTC mining profitability (PnL%) for different levels of fees defined as % of BTC mined.** Note: PnL%, calculated according to the formula 6. Hypothetical transaction fees range from 0% up to 100% of BTC mined. The market value point describes the current state of market mining efficiency assuming parameters outlined in the section Values of variables as of 2021/12/03. Source: own calculations based on data described in section Crucial assumptions and data.

Fig 26 presents the relation between energy prices, BTCUSD prices, and mining efficiency. The main observation is that the breakeven point for the mining business case is very close to the current mining condition and that it is very probable to reach it when we assume a slight increase of BTCUSD price or the use of cheaper energy from conventional or renewable energy sources. Taking into account assumed energy prices of 0.09 USD per kWh, it is very

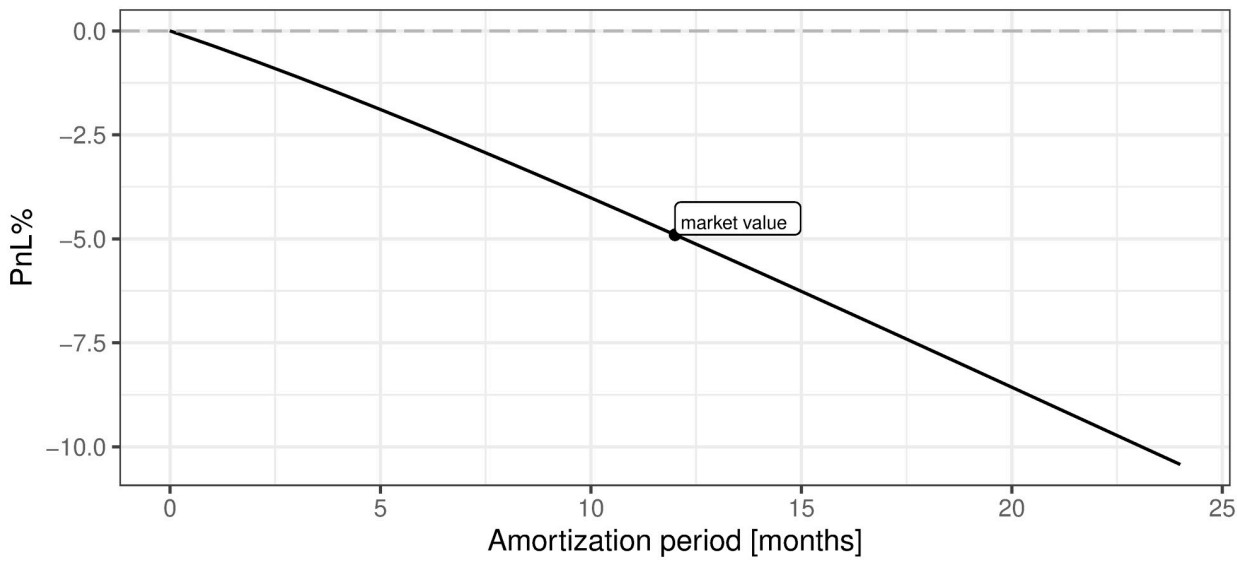

**Fig 25. BTC mining profitability (PnL%) for different levels of amortization period.** Note: PnL%, calculated according to the formula 6. Hypothetical amortization periods range from 0 to 24 months. Other parameters are assumed as outlined in the section Values of variables as of 2021/12/03. Source: own calculations based on data described in section Crucial assumptions and data.

**Table 4. Conclusions of sensitivity analysis.**

| variable | relation with mining profitability % |
|---|---|
| BTCUSD price | positive |
| Monthly hash rate increase | negative |
| Hardware total cost | negative |
| Price of energy per kWh in USD | negative |
| Transaction fee as a percentage of mining reward (Fees%) | positive |
| Amortization period | negative |

Note: The analysis assumes current mining conditions as of 2021/12/03.

possible to reach breakeven if BTCUSD prices had reached their historical highs. Therefore, after a slight change in fee structure or in hardware efficiency, the mining can become profitable under the current BTCUSD price and BTC network hash rate dynamics.

ii. Hardware cost jointly with BTCUSD price

As one can see in Fig 27 mining produces negative output in terms of PnL%. The higher BTCUSD is the more sensitive is PnL% to the total hardware cost. Nevertheless, under current assumptions of BTCUSD price PnL% hit the breakeven point if the hardware cost is on the level of 12368 USD. The problem with reaching the breakeven can be associated with a steep and sustainable increase in the BTC network hash rate while the productivity of the Antminer S19J Pro does not grow so fast. Note that even though the nominal productivity of Antminer S19J Pro is constant (104 TH/s), the relative productivity, i.e. contribution to the total network, will be only x% of its original productivity if the BTC network hash rate will increase

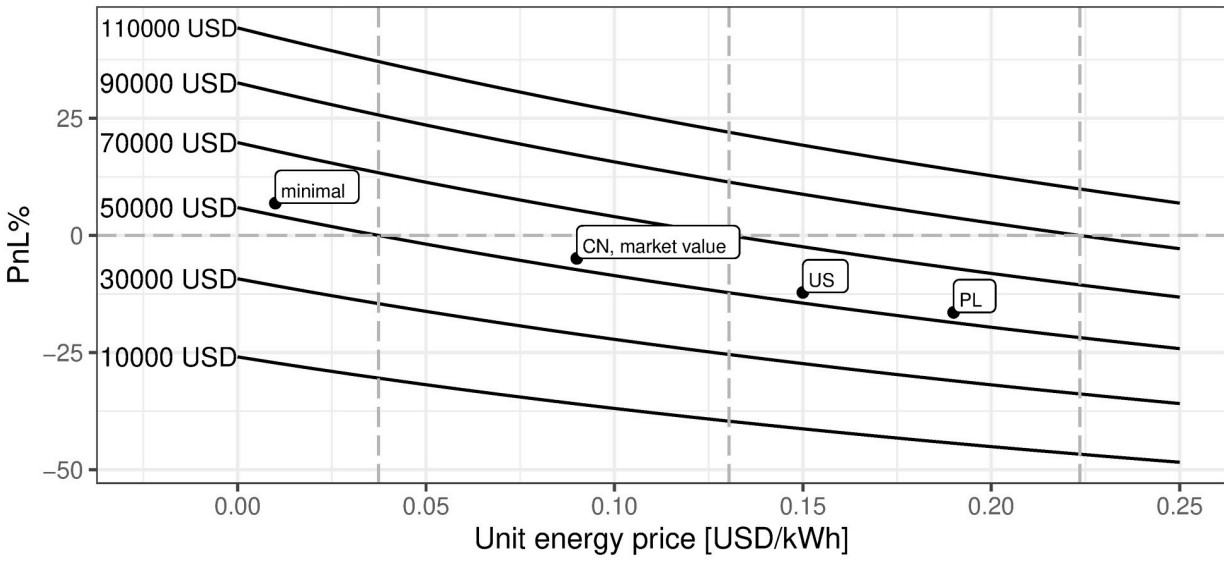

**Fig 26. BTC mining profitability (PnL%) for different levels of unit energy price and BTCUSD price.** Note: PnL%, calculated according to the formula 6. Hypothetical unit energy price ranges from 0 up to 0.25 USD per kWh and BTCUSD price ranges from 10000 USD up to 110000 USD. Minimal, CN, PL, and US define respectively minimal world energy level, China, Poland, and US located mining profitability corresponding to the local unit energy price, provided that remaining parameters are equal. The market value point describes the current state of market mining efficiency assuming the parameter outlined in the section Values of variables as of 2021/12/03, where energy cost is assumed to be 0.09 USD per kWh. Source: own calculations based on data described in section Crucial assumptions and data.

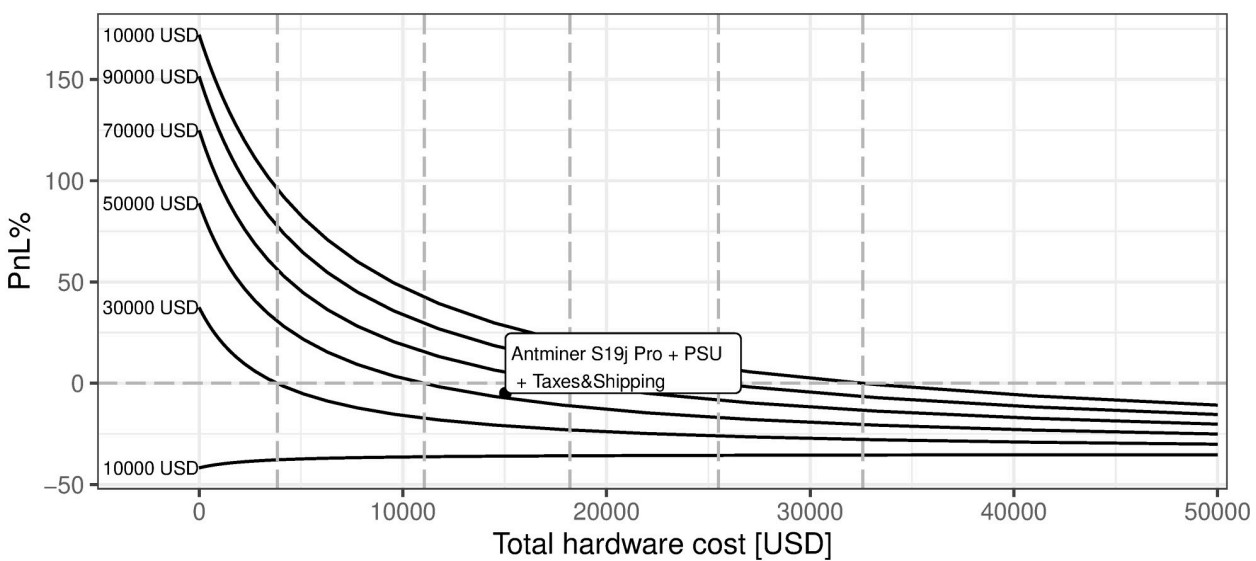

**Fig 27. BTC mining profitability (PnL%) for different levels of hardware cost and BTCUSD price.** Note: PnL%, calculated according to the formula 6. Hypothetical hardware cost per unit ranges from 0 USD up to 50000 USD and BTCUSD price ranges from 10000 USD up to 110000 USD. Antminer S19J Pro is assumed as the most optimal mining machine and its point describes the current state of market mining efficiency assuming parameters outlined in section Values of variables as of 2021/12/03. Source: own calculations based on data described in section Crucial assumptions and data.

significantly. Lower costs of the mining machine could offset the decreasing contribution to the total mining power however it will not turn the final PnL% into positive territory taking into account that the BTC network hash rate %increase will be too fast.

iii. Monthly BTC hash rate change jointly with the BTCUSD price

One of the key factors for profitability analysis is the evolution of the BTC network hash rate. The relation between this characteristic, BTCUSD price, and mining PnL% are presented in Fig 28. Assuming the current BTCUSD price, profit from mining would have appeared if the BTC network increase rate had dropped to 3.01%. Shrinking theBTC network hash rate gives hope of a stable positive PnL%. Although. theoretically one can imagine such a situation, the historical evolution of the BTC network hash rate does not give us any rationale for such expectation. Even if there were short-term periods with decreasing BTC network hash rate, the mid- and long-term trend remains increasing with an average monthly increase rate oscillating around 3.6% for the last 12 months. It will require just a small change in BTCUSD price or BTC network hash rate change% in order to make a profit on BTC mining.

iv. Fees% jointly with BTCUSD price

As it is shown in Fig 29 current market Fees% level, assumed in the analysis, do not produce enough revenue in order to make mining profitable. Worth mentioning is the fact that for the sake of integrity with other assumptions we defined the fee value at the level of its annual average. This makes the calculation less conservative in terms of PnL% due to the fact that the most recent fees values stabilize during the last few months below 2% whereas the annual average is 7.47%. Therefore, the presented results are slightly better than those obtained using the most recent Fees% level.

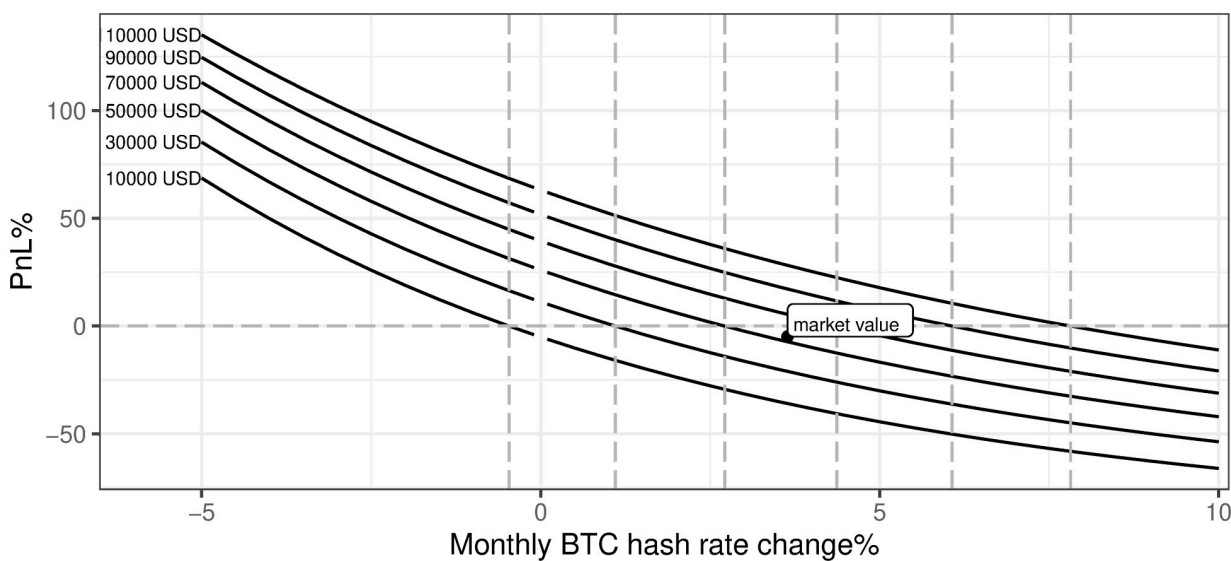

**Fig 28. BTC mining profitability (PnL%) for different evolutions of BTC network hash rate and BTCUSD price.** Note: PnL%, calculated according to the formula 6. Hypothetical monthly BTC hash rate change ranges from -5% up to 10% and BTCUSD price ranges from 10000 USD up to 110000 USD. A negative monthly BTC hash rate change means decreasing the BTC network hash rate. The market value point describes the current state of market mining efficiency assuming the parameter outlined in the section Values of variables as of 2021/12/03. Source: own calculations based on data described in section Crucial assumptions and data.

As seen in Fig 29, the possible increase in Fees% creates potential for mining to be profitable. Historically, fees have increased during periods of substantial turmoils of BTCUSD prices. We can see that the breakeven BTC %fee is reached at the level of 19.2% and the values above this number assure positive PnL% under current conditions for other variables. Therefore fee composition and accompanying drivers are the fields of current research in the crypto industry.

v. Hardware cost jointly with unit energy price

Two main conclusions from Fig 30 are that: (a) PnL% sensitivity to absolute energy price increases when hardware costs decrease and (b) PnL sensitivity to absolute hardware costs increases when energy price declines. Even though both conclusions are intuitive, the strength of dependence can not be assessed without comprehensive calculations. The results of these calculations are presented in Fig 30 so that one can easily ascertain the shape and strength of relations. Note that the subject of the analysis is a unit of Antminer S19J Pro which is believed to be the most efficient hardware under current market circumstances. The selection of the best available hardware can be seen as an additional 'degree of freedom in the model' of mining profitability. Apart from the specific parameters of a mining machine, one should consider the current market condition affecting the selection of optimal hardware. However, this subject is not a part of our research.

vi. Monthly BTC hash rate change jointly with the unit energy price

Fig 31 presents how sensitive the PnL% is to the evolution of the BTC network hash rate. Should the trend in BTC network hash rate remains unchanged i.e. 3.64% average monthly increase during the last year, the mining activity remains not profitable as long as we reach energy cost breakeven on the level of 0.054 USD per kWh (as clearly seen also in Fig 23). An

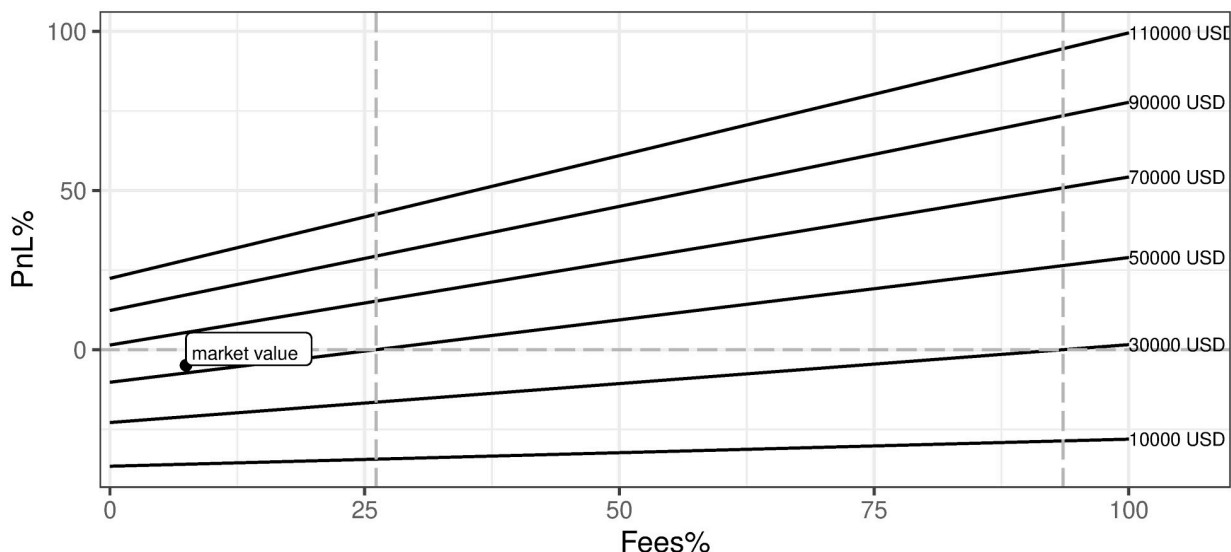

**Fig 29. BTC mining profitability (PnL%) for different levels of fees defined as % of BTC mined and different levels of BTCUSD price.** Note: PnL%, calculated according to the formula 6. Hypothetical transaction fees range from 0% up to 100% of BTC mined and BTCUSD price ranges from 10000 USD up to 110000 USD. The market value point describes the current state of market mining efficiency assuming parameters outlined in the section Values of variables as of 2021/12/03. Source: own calculations based on data described in section Crucial assumptions and data.

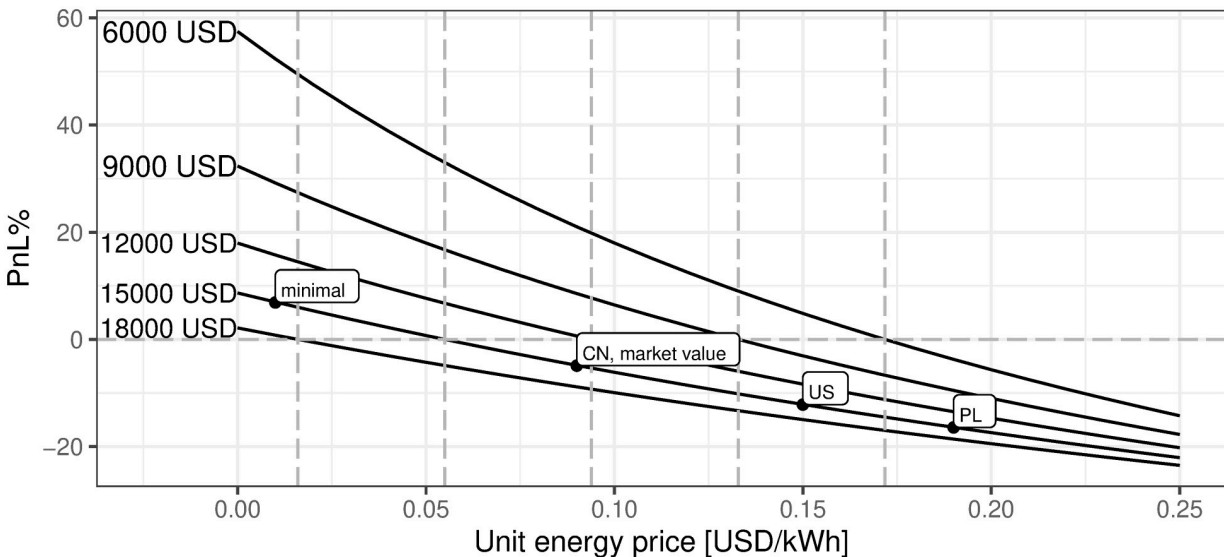

**Fig 30. BTC mining profitability (PnL%) for different levels of unit energy prices and different levels of hardware cost.** Note: PnL%, calculated according to the formula 6. Hypothetical unit energy price ranges from 0 up to 0.25 USD per kWh hardware cost per unit ranges from 6000 USD up to 18000 USD. Antminer S19J Pro is assumed as the most optimal mining machine. Minimal, CN, PL, and US define respectively minimal world energy price level, China, Poland, and US located mining profitability corresponding to the local energy price, provided that remaining parameters are equal. The market value point describes the current state of market mining efficiency assuming parameters outlined in the section Values of variables as of 2021/12/03. Source: own calculations based on data described in section Crucial assumptions and data.

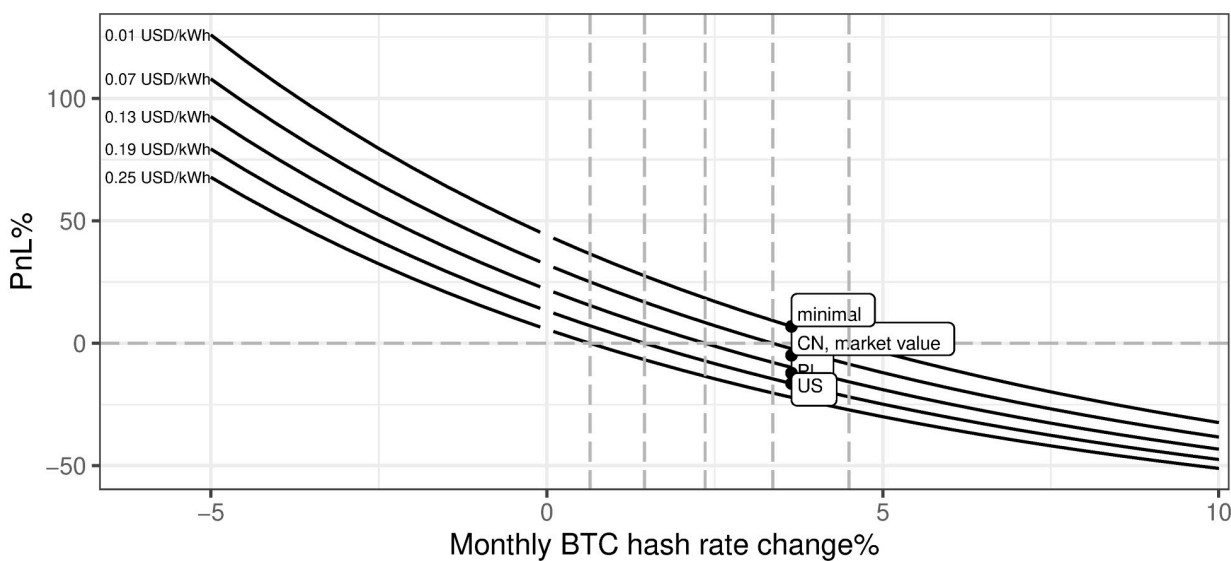

**Fig 31. BTC mining profitability (PnL%) for different evolutions of BTC network hash rate and different levels of energy prices.** Note: PnL %, calculated according to the formula 6. Hypothetical monthly BTC hash rate change ranges from -5% up to 10% and energy price ranges from 0.01 up to 0.25 USD per kWh. A negative monthly BTC hash rate change means decreasing the BTC network hash rate. Minimal, CN, PL, and US define respectively minimal world energy price level, China, Poland, and US located mining profitability corresponding to the local energy price, provided that remaining parameters are equal. The market value point describes the current state of market mining efficiency assuming parameters outlined in the section Values of variables as of 2021/12/03. Source: own calculations based on data described in section Crucial assumptions and data.

alternative interpretation of Fig 31 is that the more efficient the energy supply the more sensitive PnL% is to BTC hash rate change.

vii. Monthly BTC hash rate change jointly with hardware cost

Fig 32 shows the relation between PnL%, BTC network hash rate evolution, and hardware costs. The negative relation between PnL% and monthly hash rate change is intuitive. An interesting observation in Fig 32 is that the profitability curves ten to cross for the negative Monthly BTC hash rate change%. This means that for some scenarios of BTC network hash rate evolution the cheaper the hardware the higher the profitability, while for other scenarios the cheaper the machine the lower the profitability. Although the second part of the latter sentence seems counterintuitive, it has its economic explanation. If the network hash rate decreases the contribution of a single unit of hardware to the network will increase thus it produces more coins each consecutive day. Consequently, assuming other variables are constant, the residual value of the mining machine increases over time.

Finally, the higher the initial hardware price, the higher the profit from selling the machine at the end of the amortization period, making this component a substantial part of the revenue structure. This is of course only a theoretical consideration that is very unlikely to happen in the reality. Additionally, to get hardware residual value finite, the 12 months amortization period remains valid. Moreover, there is no chance to reach profitability just playing on hardware cost *ceteris paribus* other variables.

viii. Amortization period jointly with BTCUSD prices.

Fig 33 presents quite an interesting feature of the mining business model i.e. the dependence of profitability on the assumed amortization period for various levels of BTCUSD, with

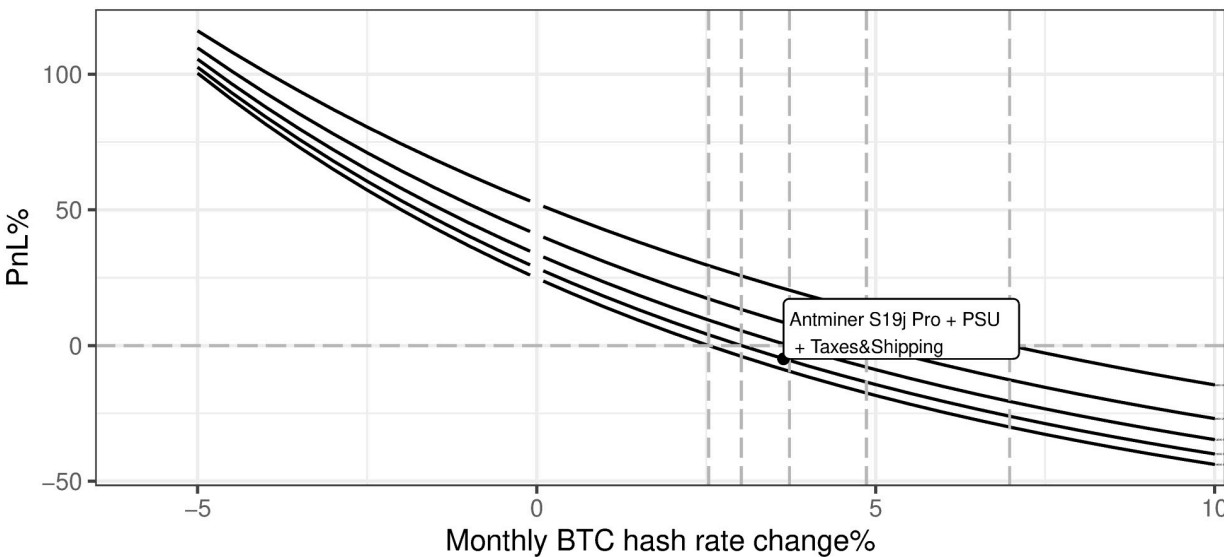

**Fig 32. BTC mining profitability (PnL%) for different evolutions of BTC network hash rate and different levels of hardware cost.** Note: PnL %, calculated according to the formula 6. Hypothetical monthly BTC hash rate change ranges from -5% up to 10% hardware cost ranges from 6000 USD up to 18000 USD. A negative monthly BTC hash rate change means decreasing BTC network hash rate. Antminer S19J Pro is assumed as the most optimal mining machine and its point describes the current state of market mining efficiency assuming parameters outlined in section Values of variables as of 2021/12/03. Source: own calculations based on data described in section Crucial assumptions and data.

other factors, held constant. One can easily see that for a higher level of BTCUSD, there is a different local maximum of profitability for various levels of BTCUSD price. This means, at least theoretically, that the most profitable scenario is actually to buy a mining machine, utilize it until the maximum profitability point and then sell the hardware out on the market, taking into account that there is another more efficient mining machine available to be bought. The

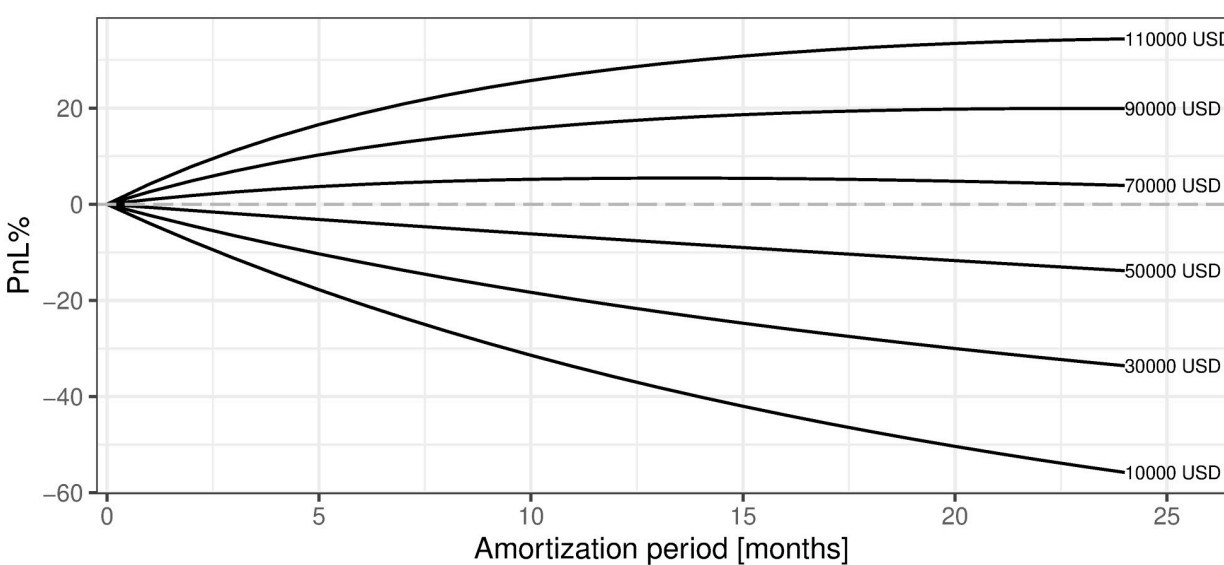

**Fig 33. BTC mining profitability (PnL%) different amortization period and different levels of BTCUSD.** Note: PnL%, calculated according to the formula 6. The hypothetical amortization period ranges from 0 up to 24 months and BTCUSD price ranges from 10000 USD up to 110000 USD. Other parameters are assumed as outlined in the section Values of variables as of 2021/12/03. Source: own calculations based on data described in section Crucial assumptions and data.

alternative scenario encompassing the full utilization of the machine until the end of the amortization period, i.e. until residual value is almost zero, produces much poorer business results. Of course, the presented situation may work assuming: (a) ceteris paribus of remaining variables, (b) an effective secondary market for trading hardware. It has to be stressed that Fig 33 will show slightly different values if one assumes different market parameters e.g. other BTC network hash rate evolution). The other conclusions should be drawn if we assume that the investor has some specific level of hurdle rate defined at the beginning.

This section, hopefully, has shed some light on the sensitivity of the mining business to the various factors and market states. The most important conclusion is that if a company wants to be efficient in this industry (if it is only possible under current conditions) it has to continuously perform multidimensional analysis on the evolution of the market, its impact on current business, its future prospects, and adjust management decisions accordingly. In some cases, it may be more profitable just to plug out mining machines and sell them out on the market.

As a summary of this part, we have to point out that the conditions under which mining starts to be profitable are nowadays very close to the current market conditions, but we would like to elaborate on this particular issue in a separate section as we regard it as one of the most important results of this paper.

## 7 Breakeven conditions for BTC mining profitability

After a detailed analysis of mining profitability and proving that BTC is not profitable under current market conditions we would like to take under consideration what actually should happen with crucial variables in order to make such activity profitable. We define the breakeven point as a state when the net mining profitability reaches zero.

The summary of breakeven values for all crucial parameters is presented in Table 5.

### 7.1 BTCUSD price

Taking into account a holistic model of BTC mining profitability i.e. covering CAPEX as well as OPEX expenses and all the sources of revenue, one may reach a conclusion about the exact BTCUSD price corresponding to the breakeven point. This level is defined at 61313 USD per BTC. This conclusion may seem to be in contradiction to information presented in several sources where so-called 'mining cost' or 'marginal cost' are defined at a far lower level. Please note that this breakeven is affected by crucial assumptions that make our research more realistic, which are: (a) the inclusion of all costs in the analysis, especially hardware costs and their amortization over time, (b) the assumption of BTC network hash rate evolution which impacts tremendously the forecasted revenue. We believe that thanks to this approach our study gained a practical aspect.

**Table 5. Summary of breakeven values for all crucial parameters.**

| Parameter | Unit | Breakeven value |
|---|---|---|
| BTCUSD price | USD | 61312.91 |
| Energy cost | USD/kWh | 0.05 |
| Hardware cost | USD | 12367.56 |
| Monthly BTC network hash rate change | % | 3.01 |
| Fees% | % | 19.21 |

Note: Breakeven values presented here are set on the level which enables %profitability of BTC mining to reach zero ceteris paribus all other variables, assuming all other conditions as listed in Table 2.

## 7.2 Unit energy price

As described in the previous section, the breakeven for energy price is currently at the level of 0.05449104 USD per kWh. Therefore, if we assume the cost of electricity is lower than the breakeven level, the output from the mining activity produces positive results. In other words —BTC mining then will cover the sum of hardware costs and non-energy related maintenance costs, assuming other factors are constant over the next 12 months. With energy cost unchanged on the level from our base case scenario, the only case when it makes sense to keep mining machines switched on is a strong conviction that the market will significantly rebound and exceed the previous highs in a short period of time.

## 7.3 Hardware cost

A similar situation as for energy cost can be observed for the hardware cost. It is possible to reach the breakeven point when the hardware cost will decrease to 12367.56 USD provided that the remaining variables are constant over the next 12 months. This means that the potential revenue will be able to cover the sum of energy and maintenance costs when the hardware cost decrease to 12367.56 USD. As a consequence one can conclude that current ASICs are not efficient enough, in terms of TH/s vs. energy consumption, to satisfy the equilibrium state defined as breakeven mining activity point.

## 7.4 Monthly BTC hash rate change%

BTC network hash rate evolution assumption is of crucial importance for mining profitability. The majority of naive web profitability calculators assume no increase in BTC hash rate while historical trends do not permit such an assumption. Therefore, after the analysis of hash rate development and its monthly increase ratio, we decided to assume a constant monthly increase ratio at the level of its last year's average value3.6%. This presumption affected our model heavily and brought it closer to the real world. A detailed analysis allowed us to define the state of the BTC hash rate dynamic that may support reaching the breakeven point at the level of 3.01% of the monthly change in the hash rate. This is more a theoretical consideration than a practical forecast as this is quite low value for a longer period of time.

## 7.5 Fees%

The transaction fee is the area of probable future adjustments if BTC mining is to regain profitability. Even though we do not place this part in the center of our research, we can derive a rough conclusion on the fee level corresponding to the breakeven point ceteris paribus. The mining would have turned into profitable territory if the fee had increased to 19.2% of BTC mined, i.e. additionally 1.2BTC per block. This area remains of interest to developers and scientists s, especially in the context of halving reward day which lastly happened in 2020.

## 8 Conclusions

The main important contribution of this paper is the verification of BTC mining profitability with regard to the current state of the BTC network and the cost of various energy sources. As far as we know, it is the first such thorough attempt supported by a robustness check with regard to all crucial parameters of this process. The research we have presented allows us to tackle our main hypothesis and additionally to answer several research questions. With no doubt, we can refer to our main research hypothesis (RH1): *Is BTC mining profitable under current mining conditions?* Under current market conditions mining is not profitable but is very close to being profitable. Detailed calculations which enable us to formulate such answers

are presented in previous sections. The problem is that such an answer raises many other questions connected with the overall blockchain and cryptocurrency environment which require further research. Below we will refer to all of them one by one.

- RQ1: What is the sensitivity of BTC mining profitability to initial assumptions and future trajectories of main parameters?

There is a strong positive correlation between BTC mining profitability and BTCUSD and Fees%. Simultaneously there is negative sensitivity of BTC mining profitability to the monthly BTC hash rate change, energy price, hardware cost, and amortization period. Even though the relations are in most cases intuitive, there are several interesting, and sometimes surprising, conclusions like the non-monotonic relation of PnL% and amortization period or hardware cost. They are presented in detail in the section Sensitivity Analysis (Section 6).

- RQ2: Is there a strong positive relation between BTCUSD price and the difficulty/hash rate of the BTC network?

Based on the presented analysis one can say there is low positive relation between BTCUSD price and difficulty/hash rate of the BTC network. However, in the recent months of the 2021 year the relationship started to be much stronger.

- RQ3: What are the breakeven levels of the main BTC mining parameters in order to make this activity profitable?

Assuming ceteris paribus all other variables one can conclude that 61312.91 USD per BTC or an increase of BTC Fees% to 19.2% turns mining into a profitable zone. Alternatively, either a decrease in the average monthly BTC network change to 3.01% or energy cost to 0.0545 USD per kWh or the hardware cost to 12367.6 USD may satisfy the breakeven point. More theoretical considerations are presented in the section Main Results (Section 5).

- RQ4: What is the rational level of the amortization period for our mining machines?

Since the mining is currently not profitable the answer is 0 months. However, for various scenarios of the market evolution, the optimal amortization period may be on different levels. One of the conclusions sis that miners should consider utilizing the hardware up to some point and then sell out the machine instead of mining until the full utilization of the machine. This approach may increase business results and is explained in section b of the Main results.

- RQ5: What are the consequences of BTC mining efficiency for the future ability of blocks to be mined further?

We present six theoretical scenarios being an attempt to refer to RQ5, nevertheless one may imagine a mix of these scenarios pushing BTC mining toward the breakeven point. In scenarios from one to five, we assume that the BTC network will still exist and is able to confirm blocks and evolve while in the last one, we take into account that current mining unprofitability could weaken and destroy the network as a whole.

- RQ6: Can we expect that blocks will not be confirmed if the efficiency of BTC mining will be negative for an extended period of time?

This is one of the most important questions in this paper because this issue affects not only single mining endeavors but can influence the probability of the BTC network to work properly as a whole. The answer is NO, at least for a longer period of time. Blocks will be mined unless there is trust in the BTC blockchain protocol, regardless of the BTCUSD price and other variables. Since the BTC ecosystem has its own immune mechanism that helps to sustain

the protocol healthy, we assume that appropriate adaptation will take place to bring the BTC blockchain to the equilibrium state in the long term.

- RQ7: Should we invest in BTC mining if we expect very strong growth of BTC price?

The answer is definitely NO, unless you are a hardware mining producer thus having a synergy effect. The better investment option for a potential miner/investor is to turn toward the BTCUSD spot or derivative market, and open a position strictly dependent on an increase in the price of BTC. In such a way, the potential investor can gain from a 'pure' BTC price increase while such a trade will not depend on other characteristics specific to BTC mining activity, like energy and hardware costs, BTC network hash rate, or BTC network difficulty.

- RQ8. Is it possible that technological advancements in ASIC's performance would change the BTC mining landscape and make mining profitable?

The introduction of faster ASIC chips will definitely change the BTC mining landscape, at least for the company that will be the first to introduce the new hardware. Mainly because the most likely scenario is that such a company will leverage on the new technology for its own mining activity and then supply ASICs to the market. The additional factor is the price of new hardware.

- RQ9: What kind of new financial instruments may stimulate the BTC mining industry?

Introduction of futures or other derivatives for the monthly (quarterly, yearly, etc.) BTC network hash rate change could stimulate the mining industry, creating an opportunity to stabilize performance and hedge against adverse scenarios. Therefore it may strengthen BTC as a digital currency ecosystem.

Even though our paper is relatively long, what was caused by the fact that we tried to refer to the subject of BTC mining profitability with the highest scientific and practical rigidity at the end we found that there are still several issues that should be resolved especially when we treat the subject of cryptocurrency mining more broadly. These questions can be summarized as follows:

- What is the mining profitability of other cryptocurrencies, the main ones?

- Proof-of-stake versus Proof-of-work. Should we go this way in the case of the BTC network?

- What is the future shape of transaction fees?

- The reward halving challenge and its consequences.

- To prepare an analysis of historical BTC mining conditions in order to answer the question if the undertaking of BTC mining activity was reasonable in the past.

We are sure that they can be treated as a subject of future scientific investigations or practical reports.

## Supporting information

**S1 File. This file contains the necessary data to reproduce the results presented in the study.**
(CSV)

## Author Contributions

**Conceptualization:** Krzysztof Kosc, Przemysław Ryś, Paweł Sakowski, Robert Ślepaczuk, Grzegorz Zakrzewski.

**Data curation:** Małgorzata Jabłczyńska, Krzysztof Kosc, Paweł Sakowski, Grzegorz Zakrzewski.

**Formal analysis:** Małgorzata Jabłczyńska, Paweł Sakowski, Robert Ślepaczuk, Grzegorz Zakrzewski.

**Investigation:** Małgorzata Jabłczyńska, Krzysztof Kosc, Grzegorz Zakrzewski.

**Methodology:** Krzysztof Kosc, Paweł Sakowski, Robert Ślepaczuk, Grzegorz Zakrzewski.

**Resources:** Przemysław Ryś, Grzegorz Zakrzewski.

**Software:** Małgorzata Jabłczyńska, Paweł Sakowski, Grzegorz Zakrzewski.

**Supervision:** Robert Ślepaczuk.

**Validation:** Małgorzata Jabłczyńska, Paweł Sakowski, Robert Ślepaczuk.

**Visualization:** Przemysław Ryś, Paweł Sakowski, Robert Ślepaczuk.

**Writing – original draft:** Przemysław Ryś, Robert Ślepaczuk, Grzegorz Zakrzewski.

**Writing – review & editing:** Krzysztof Kosc, Paweł Sakowski, Robert Ślepaczuk, Grzegorz Zakrzewski.

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
