## [Decision Letter · Decision Letter 0]

6 Jan 2023

PONE-D-22-34332Energy and cost efficiency of bitcoin mining endeavorPLOS ONE

Dear Dr. Ślepaczuk,

Thank you for submitting your manuscript to PLOS ONE. After careful consideration, we feel that it has merit but does not fully meet PLOS ONE’s publication criteria as it currently stands. Therefore, we invite you to submit a revised version of the manuscript that addresses the points raised during the review process.

The analysis made by the authors, taking into account the judgment of the reviewers, although interesting, needs considerable improvements.

In this regard, please take into account all the comments of the authors.

We look forward to receiving your revised manuscript.

Kind regards,

Antonio Luciano Martire

Academic Editor

PLOS ONE

Journal Requirements:

Reviewers' comments:

Reviewer's Responses to Questions

**Comments to the Author**

1. Is the manuscript technically sound, and do the data support the conclusions?

Reviewer #1: Yes

Reviewer #2: Partly

Reviewer #3: Yes

2. Has the statistical analysis been performed appropriately and rigorously? 

Reviewer #1: Yes

Reviewer #2: N/A

Reviewer #3: Yes

3. Have the authors made all data underlying the findings in their manuscript fully available?

Reviewer #1: Yes

Reviewer #2: Yes

Reviewer #3: Yes

4. Is the manuscript presented in an intelligible fashion and written in standard English?

Reviewer #1: No

Reviewer #2: No

Reviewer #3: Yes

5. Review Comments to the Author

Reviewer #1: I would like to thank the editor for an opportunity review this manuscript. This paper investigates the efficiency of BTC mining under current market conditions (December 2021) including soaring energy prices produced from many different sources in different geographical locations. Overall, this study makes contributions to the current literature. I am overall positive regarding the work, which is why I would be open to reviewing a revised manuscript; however the revisions are major. Below presented some detailed comments. Hopefully they can assist the authors to improve the quality of the paper.

1.The Introduction should further motivate the study. Why this study is necessary? What policy level problem this study is addressing? How the study is expected to provide any solution to that problem? How the choice of sample is complementing that problem? Are the results and policies generalizable? Introduction is silent in all these aspects. Mere choice of new variables, new methods, or choosing a new context are not considered as contribution of a study. Try to find out the policy level problem, as academic literature will not be able to provide you with the specific policy issue.

2.In the introduction and literature review, you need to discuss the existing literature to improve the motivation of the paper and broaden the view of readers and display the contribution of this paper. You may consider the following papers.

https://doi.org/10.1016/j.energy.2022.124172;https://doi.org/10.1016/j.rser.2022.113058

3.What is the aim of the review of literature? The authors have merely listed out the studies without even creating a debate among them. Without that debate and thoughtful contradictions, the research gap cannot be substantiated.

4.Why should the authors use this method? You need to provide more comparison analysis with other methods. Why should the authors use this method? You need to provide more analysis about the economic reasoning.

5.The tables are not clear and the authors should add the notes under each table to explain the meanings.

6.Empirical findings – please link your findings more strongly to the: (i) theory, (ii) empirics, (iii) context; and (iv) highlight their economic, academic/research and policy implications. Closely link up and cite the papers that you have discussed in the background, theory and empirical literature review & and hypotheses development section to the findings you are presenting here.

7.More robust analysis should be provided.

8.Conclusion – Please outline a summary of findings, contributions, implications, limitations and avenues for future research. Especially, expand the discussions relating to implications, limitations and avenues for future research.

9.Finally, this manuscript needs careful editing by someone with expertise in technical English editing paying particular attention to sentence structure so that the goals and results of the study are clear to the readers. Please there are considerable number of typos, spelling errors and grammatical mistakes throughout the paper that a careful reading will help you to eliminate them.

Reviewer #2: The article analyzes the profitability of Bitcoin mining activities. As a result, it is shown that they are not profitable or only profitable in exceptional cases. Overall, however, the article has significant shortcomings. It is neither well structured nor is the research contribution sufficiently clear. The following suggestions should be considered when revising the article:

1. The research questions should be better summarized and better structured. This would also improve the reading flow later in the article.

2. The additional insight gained from the article ultimately remains unclear and should be pointed out precisely. After the literature review, the article merely summarizes succinctly: "The point is that none of these works covered the subject of BTC mining efficiency thoroughly enough in order to refer to all our hypotheses and research questions and this was the reason why we decided to undertake this topic."

3. In the third chapter, a number of metrics are introduced without deeper reflection. It remains unclear how they relate to the overall model. Even more critically, descriptive data for individual metrics are reported and discussed. Thus, there is no clear separation between reflective introduction of the methodology, documentation of results and the discussion of the article's contributions. Even more surprising is the fact that the fourth chapter introduces the methodology, which further complicates the readability of the article. The structure of the article should therefore be comprehensively revised and adapted to scientific standards.

4. The role of numerous variables and their calculation remains unclear. This applies, for example, to transaction cost fees, which are mentioned in rudimentary form in several places but never discussed in more detail. As a result, many parts of the article seem strangely fragmented and a good reading flow is hardly possible.

5. It is at least surprising that neither the energy mix (only different cost levels) nor environmental aspects of Bitcoin mining are addressed in more depth.

6. In Figure 17, the light gray bars are obviously interchanged with the dark gray bars in the legend. Otherwise, there would be a clear contradiction to the documented results.

7. Overall, the article is in great need of revision, both in terms of content and linguistic quality. I suggest that a clearer structure including the streamlining of the research questions will be created, that each section should is revised with regard to its contribution to the content, and that the overall precision of the explanations will be significantly increased. I would also suggest to drop some of the figures as they do not add value to the research outcome of the article. Finally, the paper should be thoroughly proofread by a native speaker to eliminate expression and grammatical weaknesses.

I wish the authors success in revising the paper.

Reviewer #3: The work is just a descriptive analysis article for gathered data about mining process, nothing exceptional to be published in this journal . Although the authors try to get close to the reality, the “current conditions of BTC mining (as of 2021/12/03) in the main worldwide jurisdiction” make the whole analysis just a mirror of the situation in that specific date where the base case scenario assumes a value of BTC of 53671.61 USD. This creates a distortion on the perception of the value of the BTC because of the drawdown of the last year with the current price 16,790.40 USD, so this makes this paper difficult for forecasting predictions. The authors clarify at the break even conditions the BCTUSD price where the break even value is 61312.91 USD, which in this level was reached in October 2021 and stayed for one month above that level in the overall history of bitcoin.

The question is, if we produce bitcoin by mining, which is a very slow process, can we afford the cost to produce them and wait until we reach the requested level in order to sell?

Another condition, at ceteris paribus with the other, is the energy cost, from the tittle of the article, it is easy to understand which are the states have a low kWh price, with the break even decided at level of 0.05 kWh, but there no comparable analysis with other models of production of the bitcoins. The states have adapted the jurisdiction in different ways, so we cannot make the same assumptions.

The sensitivity analysis is done by changing one variable, at ceteris paribus or, by changing the other two which is practically fixing the level of one variable and estimating the other but there is a need to compare the results with another paper.

I propose the following corrections:

1. Study the correlation between BTCUSD , energy cost, hardware cost and the other parameters all together, to understand their impact in the PnL% by creating a model and study their elements.

2. Compare the results with other authors in blockchain industry, perhaps with different cryptocurrency like Ethereum.

3. Clarify the answer to RQ7 at the conclusion.

4. RQ8: The hardware cost changes from states to state, and there is a need to compare between different machines.

5. RQ9, the options market for cryptocurrency is out of the purpose of the paper.

6. PLOS authors have the option to publish the peer review history of their article (what does this mean?). If published, this will include your full peer review and any attached files.

Reviewer #1: No

Reviewer #2: No

Reviewer #3: **Yes: **Denis Veliu

---

## [Author Response · Author response to Decision Letter 0]

23 Feb 2023

Dear Editor and Reviewers,

Our responces to Editor and Reviewers comments are placed in the separate document attached to this submission named: "2023-02-17_Response-to-Reviewers_PlosONE.pdf"

Kind regards,

Robert Ślepaczuk

---

## [Decision Letter · Decision Letter 1]

14 Mar 2023

Energy and cost efficiency of bitcoin mining endeavor

PONE-D-22-34332R1

Dear Authors,

We’re pleased to inform you that your manuscript has been judged scientifically suitable for publication and will be formally accepted for publication once it meets all outstanding technical requirements.

Kind regards,

Antonio Luciano Martire

Academic Editor

PLOS ONE

Additional Editor Comments (optional):

I have carefully read the article considering the modifications requested by two reviewers who propose acceptance. The authors have, also in my opinion, satisfied all the requests. 

Reviewers' comments:

Reviewer's Responses to Questions

**Comments to the Author**

1. If the authors have adequately addressed your comments raised in a previous round of review and you feel that this manuscript is now acceptable for publication, you may indicate that here to bypass the “Comments to the Author” section, enter your conflict of interest statement in the “Confidential to Editor” section, and submit your "Accept" recommendation.

Reviewer #1: All comments have been addressed

Reviewer #2: (No Response)

Reviewer #3: All comments have been addressed

2. Is the manuscript technically sound, and do the data support the conclusions?

Reviewer #1: Yes

Reviewer #2: Partly

Reviewer #3: Yes

3. Has the statistical analysis been performed appropriately and rigorously? 

Reviewer #1: Yes

Reviewer #2: N/A

Reviewer #3: Yes

4. Have the authors made all data underlying the findings in their manuscript fully available?

Reviewer #1: Yes

Reviewer #2: Yes

Reviewer #3: Yes

5. Is the manuscript presented in an intelligible fashion and written in standard English?

Reviewer #1: Yes

Reviewer #2: No

Reviewer #3: Yes

6. Review Comments to the Author

Reviewer #1: (No Response)

Reviewer #2: The authors have only cosmetically revised the article and have hardly addressed the critical comments of all reviewers in depth. With reference to my review, this applies to comments 1-5. For example, with reference to my comment #5, the topic of the energy mix has not been addressed at all. There is only a brief reference to the carbon footprint in the literature review. However, this important issue is not followed up in any of the calculations. This problem should at least be addressed quantitatively (1) in the sensitivity analyses.

Ultimately, this also applies to my comment #7. Please take also a look at the structure of the article and the section (e.g. 3.1-3.14!, number of research questions), which often contain only one sentence. We would not even accept this from our students. As I wrote a thorough revision needs a better structure of the entire article.

Overall, I recommend to better address the comment 1-5 and 7. Only comment 6 (referring to the original figure 17) is resolved. In this state I still do not recognize a sufficient scientific contribution to the existing literature.

Reviewer #3: The requested changes have been made and the questions are sufficiently and exhaustive answered. The work still have some difficult condition to be satisfied in order that the production of bitcoin can be suitable. Thus, since last year it was the window of oppurtunity in which these conditions were satisfied, we can say that in can be applied in real case scenario.

Best regards

7. PLOS authors have the option to publish the peer review history of their article (what does this mean?). If published, this will include your full peer review and any attached files.

Reviewer #1: No

Reviewer #2: No

Reviewer #3: No

---

## [Editor Report · Acceptance letter]

20 Mar 2023

PONE-D-22-34332R1 

Energy and cost efficiency of Bitcoin mining endeavor  

Dear Dr. Ślepaczuk:

I'm pleased to inform you that your manuscript has been deemed suitable for publication in PLOS ONE. Congratulations! Your manuscript is now with our production department. 

Kind regards, 

on behalf of

Dr. Antonio Luciano Martire 

Academic Editor

PLOS ONE